# *Prosopis* Plant Chemical Composition and Pharmacological Attributes: Targeting Clinical Studies from Preclinical Evidence

**DOI:** 10.3390/biom9120777

**Published:** 2019-11-25

**Authors:** Javad Sharifi-Rad, Farzad Kobarfard, Athar Ata, Seyed Abdulmajid Ayatollahi, Nafiseh Khosravi-Dehaghi, Arun Kumar Jugran, Merve Tomas, Esra Capanoglu, Karl R. Matthews, Jelena Popović-Djordjević, Aleksandar Kostić, Senem Kamiloglu, Farukh Sharopov, Muhammad Iqbal Choudhary, Natália Martins

**Affiliations:** 1Phytochemistry Research Center, Shahid Beheshti University of Medical Sciences, Tehran 1991953381, Iran; 2Department of Medicinal Chemistry, School of Pharmacy, Shahid Beheshti University of Medical Sciences, Tehran 11369, Iran; 3Department of Chemistry, Richardson College for the Environmental Science Complex, The University of Winnipeg, Winnipeg, MB R3B 2G3, Canada; a.ata@uwinnipeg.ca; 4Department of Pharmacognosy and Biotechnology, School of Pharmacy, Shahid Beheshti University of Medical Sciences, Tehran 11369, Iran; 5EvidenceBased Phytotherapy & Complementary Medicine Research Center, Alborz University of Medical Sciences, Karaj 19839-63113, Iran; nafiseh.khosravi85@gmail.com; 6Department of Pharmacognosy, School of Pharmacy, Alborz University of Medical Sciences, Karaj 19839-63113, Iran; 7G. B. Pant National Institute of Himalayan Environment and Sustainable Development, Garhwal Regional Centre, Upper Baktiyana, Srinagar-246 174, Uttarakhand, India; arunjugran@gbpihed.nic.in; 8Faculty of Engineering and Natural Sciences, Food Engineering Department, Istanbul Sabahattin Zaim University, Halkali, 34303 Istanbul, Turkey; mervetomas@gmail.com; 9Faculty of Chemical and Metallurgical Engineering, Food Engineering Department, Istanbul Technical University, Maslak, 34469 Istanbul, Turkey; capanogl@itu.edu.tr; 10Department of Food Science, Rutgers University, New Brunswick, NJ 08901-8520, USA; bijan@sebs.rutgers.edu; 11Faculty of Agriculture, Chair of Chemistry and Biochemistry, University of Belgrade, 11080 Belgrade, Serbia; jelenadj@agrif.bg.ac.rs (J.P.-D.); kosticmeister@gmail.com (A.K.); 12Mevsim Gida Sanayi ve Soguk Depo Ticaret A.S. (MVSM Foods), Turankoy, Kestel, 16450 Bursa, Turkey; senemkamiloglu87@gmail.com; 13Department of Pharmaceutical Technology, Avicenna Tajik State Medical University, Rudaki 139, Dushanbe 734003, Tajikistan; shfarukh@mail.ru; 14H.E.J. Research Institute of Chemistry, International Center for Chemical and Biological Sciences, University of Karachi, Karachi 75270, Pakistan; iqbal.choudhary@iccs.edu; 15Faculty of Medicine, University of Porto, Alameda Prof. Hernâni Monteiro, 4200-319 Porto, Portugal; 16Institute for Research and Innovation in Health (i3S), University of Porto, 4200-135 Porto, Portugal

**Keywords:** *Prosopis*, vitexin, C-glycosyl flavones, food preservative, antiplasmodial, wound healing potential

## Abstract

Members of the *Prosopis* genus are native to America, Africa and Asia, and have long been used in traditional medicine. The *Prosopis* species most commonly used for medicinal purposes are *P. africana*, *P. alba*, *P. cineraria*, *P. farcta*, *P. glandulosa*, *P. juliflora*, *P. nigra*, *P. ruscifolia* and *P. spicigera*, which are highly effective in asthma, birth/postpartum pains, callouses, conjunctivitis, diabetes, diarrhea, expectorant, fever, flu, lactation, liver infection, malaria, otitis, pains, pediculosis, rheumatism, scabies, skin inflammations, spasm, stomach ache, bladder and pancreas stone removal. Flour, syrup, and beverages from *Prosopis* pods have also been potentially used for foods and food supplement formulation in many regions of the world. In addition, various in vitro and in vivo studies have revealed interesting antiplasmodial, antipyretic, anti-inflammatory, antimicrobial, anticancer, antidiabetic and wound healing effects. The phytochemical composition of *Prosopis* plants, namely their content of C-glycosyl flavones (such as schaftoside, isoschaftoside, vicenin II, vitexin and isovitexin) has been increasingly correlated with the observed biological effects. Thus, given the literature reports, *Prosopis* plants have positive impact on the human diet and general health. In this sense, the present review provides an in-depth overview of the literature data regarding *Prosopis* plants’ chemical composition, pharmacological and food applications, covering from pre-clinical data to upcoming clinical studies.

## 1. Introduction

Medicinal plants have been used since the beginning of human civilization to treat various diseases. Different properties have been discovered for each plant, due to many researchers’ focus on plants as a natural resource for treating human health [1,2]. Of the various medicinal plants, species belonging to the *Prosopis* genus have been widely used in folk medicine. The *Prosopis* genus belongs to the Fabaceae or Leguminosae family, and includes about 45 species of spiny trees and shrubs. This genus is found in both subtropical and tropical areas of the world. Briefly, species belonging to the *Prosopis* genus have been traditionally used for the treatment of asthma, birth/postpartum pains, callouses, conjunctivitis, diabetes, diarrhea, expectorant, fever, flu, lactation, liver infection, malaria, otitis, pains, pediculosis, rheumatism, scabies, skin inflammations, spasm, stomach ache, removal of bladder and pancreas stones, among other applications [3,4,5]. On the other hand, in addition to being being used for centuries for medicinal purposes *Prosopis* plants are also of commercial interest. The paste, gum, and leaves and pods smoke of *Prosopis* plants possess various bioactive properties, such as anticancer, antidiabetic, anti-inflammatory, antimicrobial and antioxidant effects [6,7,8]. These effects have been increasingly correlated with their content in phytoconstituents, namely flavonoids, tannins, alkaloids, quinones and phenolics. Indeed, *Prosopis* plants have been reported as a rich source of phenolic compounds, being anthocyanins and the flavonoids apigenin, luteolin, quercetin and their derivatives the most abundant ones [9,10,11].

Thus, and given the above highlighted aspects, the aim of this review is to provide an in-depth overview of the literature data on the biological activities of the *Prosopis* genus, and to present its potential benefits and applications in both the pharmaceutical and food industries.

## 2. *Prosopis* Plants Phytochemical Composition

Epidemiological studies have suggested an inverse association between the consumption of phytochemicals (such as carotenoids and phenolics) and a reduced risk of certain diseases, namely chronic disorders [12]. Medicinal plants, and specifically *Prosopis* plants, are rich sources of phytochemicals, among them alkaloids, phenolic compounds, particularly flavonoids (Figure 1; Figure 2) and phenolic acids (Figure 3), glycosides, steroids, tannins and triterpenoids, increasingly recognized as having positive health effects.

Several studies have been conducted to identify and quantify the chemical composition of *Prosopis* plants. The most commonly studied plants and their abundance in phytochemicals are listed in Table 1.

## 3. Traditional Medicinal Uses of *Prosopis* Plants

Recent ethnopharmacological studies have shown that *P. africana*, *P. alba*, *P. cineraria*, *P. farcta*, *P. glandulosa*, *P. juliflora*, *P. nigra*, *P. ruscifolia* and *P. spicigera* are amongst the most commonly used *Prosopis* plants in folk medicine treatments. As shown in Table 2, different *Prosopis* plants parts are used in diverse regions of the world for the treatment of various diseases. In particular, the positive health effects of *P. cineraria*, *P. juliflora* and *P. africana* have been well-documented [3,4,5,13,14,15,16,17,18,19,20,21,22].

### 3.1. Prosopis cineraria

*P. cineraria* is traditionally used as a medicine in different regions of Pakistan, including Bahawalnagar in Punjab Province, the Thar Desert (Sindh) and Hafizabad district in Punjab (Table 2). The oral or topical administration of *P. cineraria* leaves, stems, fruits, flowers, barks and pods is used for the treatment of spasms, diabetes, liver infection, diarrhea, bladder and pancreas stones, fever, flu, rheumatism, leucorrhea, boils, blisters, scorpion bite, chronic dysentery, cataract, asthma, sexually-transmitted infections, and gynecological complaints, including menstrual disorders, as contraceptive and to prevent abortion [3,4,5,13,15,16]. Besides Pakistan, in the South of Kerman in Iran, *P. cineraria* flowers are also topically applied to treat asthma and skin rashes [14].

### 3.2. Prosopis juliflora

*P. juliflora* is commonly used in Pakistan, and can be either orally consumed or topically applied. Leaves, gum, whole plant, flower, stem and bark of *P. juliflora* are used as a painkiller, body tonic, galactagogue, and expectorant, or to treat boils, eye inflammation, muscular pain, kidney stones, toothache, breast cancer, asthma and cough in Thar Desert (Sindh), Bahawalnagar in Punjab province and Mohmand Agency of Federally Administered Tribal Areas (FATA) [3,4,13,17]. In addition to Pakistan, the stem bark of *P. juliflora* is also consumed in western Madhya Pradesh, India for the treatment of asthma [18].

### 3.3. Prosopis africana

*P. africana* is commonly known as African mesquite, and is also traditionally used as a medicine. The oral administration of leaves and bark of this plant is used for the treatment of malaria in Sélingué subdistrict in Mali [19] and Nsukka Local Government Area in south-eastern Nigeria [21].

In addition to malaria, various parts of *P. africana*, including roots, leaves and bark, are also used as analgesic and anti-inflammatory in Guinea-Bissau [22] and for the treatment of pains, pregnancy-related conditions (childbirth, breastfeeding, newborn diseases), and skin inflammations (i.e., wounds, burns) in north-west of Nigeria [20].

### 3.4. Other Prosopis Plants

The resin of *P. alba* and *P. nigra*, and the leaves of *P. ruscifolia* have also been consumed by the Wichí people of Salta province in Argentina against conjunctivitis, post-abortion pain, ocular trauma, stomachache, pimples or rash, scabies, callouses, fever, birth or postpartum pains, diarrhea, pediculosis and otitisSuárez [23].

A study carried out in South America (Bustamante, Nuevo León, Mexico), also pointed out that the ingestion of *P. glandulosa* inflorescences may be useful to relief stomach pain [25]. In addition to the above, in Jahrom, Iran, *P. farcta* fruits have been consumed to prevent constipation and to reduce fever [24], whereas *P. spicigera* bark, leaves and flowers are used for the treatment of asthma in Pakistan [4].

## 4. Biological Activities *Prosopis* Plants

Increasingly renowned bioactivities have been attributed to *Prosopis* plants, formerly assessed from pre-clinical (in vitro and in vivo studies) and more recently validated by a raising number of clinical studies, supporting their folk medicinal uses. In the following sections, the in vitro and in vivo bioactive effects of the most widely recognized *Prosopis* plants are described, and the clinical studies performed so far are also the focus of attention. There have been a wide variety of pre-clinical studies performed using *Prosopis* plants to assess their bioactive effects. The most representative ones carried out in vitro are briefly summarized in Table 3 and discussed below.

### 4.1. In vitro Biological Effects of Prosopis Plants

#### 4.1.1. *Prosopis juliflora*

##### Antioxidant Effects

The ethanol extract of *P. juliflora* leaves was investigated for antioxidant activity, revealing interesting radical scavenging activities (RSA) even more evident than the control propyl gallate [26].

##### Antiplasmodial Effects

The antiplasmodial potential of *P. juliflora* ethanol extract was studied at varying doses of filter- sterilized extracts (100, 50, 25, 12.5, 6.25 and 3.125 μg/mL), against *Plasmodium falciparum*. *P. juliflora* leaf, bark and flower extracts exhibited and inhibitory concentration (IC_50_) values >100 μg/mL. A time- and dose-dependent effect was also stated. Further, no chemical injury was detected in the erythrocytes incubated with all the ethanolic extracts of the species. The in vitro antiplasmodial property appeared to be associated with the content of alkaloids, glycosides, carbohydrates, flavonoids, phenols, saponins, triterpenoids, proteins and tannins in the tested ethanolic extracts [27].

##### Antimicrobial Effects

*P. juliflora* leaves methanolic extract was also studied against seven Gram negative (*Escherichia coli*, *E. coli* Extended Spectrum β-Lactamase (ESBL), *Shigella flexneri*, *Salmonella typhi*, *Proteus mirabilis*, *Pseudomonas aeruginosa*, and *Klebsiella pneumoniae*) and three Gram positive (*Enterococcus faecalis*, *Listeria monocytogenes* and *Bacillus cereus*) bacteria. All bacteria screened were revealed to be sensitive to the methanolic extract (inhibition zone varied from 12 to 41 mm), although more prominent activity was reported against *P. aeruginosa* (41 mm) and *L. monocytogenes* (33 mm) at a dose of 100 mg/mL [28]. *P. juliflora* seed pod methanol extract also demonstrated antibacterial effects against *S. aureus* (0.312 mg/mL), *Staphylococcus epidermidis* (0.078 mg/mL), and for both *E. coli* and *P. aeruginosa* (1.25 mg/mL) [29]. Similar findings were also stated by other authors, were the methanol extract was effective against *Staphylococcus aureus*, *Micrococcus luteus*, *Bacillus cereus*, *Shigella sonee*, *P. aeruginosa* and *E. coli*, with varying degrees of sensitivity also being observed. *P. aeruginosa* was the most resistant and *M. luteus* had the less resistance [30]. *P. juliflora* was also reported as effective against *Neisseria gonorrhoeae* [31]. *P. juliflora* leaves extract was also studied against ten bacterial cultures, where the maximum antibacterial effects were found for aqueous fractions as compared to solvent fractions [32]. The dose-dependent antibacterial effect of silver nanoparticles (AgNPs) from the aqueous extract of *P. juliflora* leaf revealed good effects against *E. coli* and *P. aeruginosa* [33]. Similarly, the antifungal activity of different extracts (aqueous, petroleum ether, benzene, chloroform, methanol and ethanol extracts) and alkaloid extract of *P. juliflora* leaves was studied against *Alternaria alternata*. Aqueous extract exhibited significant antifungal activity at 24% concentration. Methanol and ethanol extract displayed significantly higher antifungal effects, when compared with the other solvent extracts used. Fractionation of the methanol extract led to the isolation of an alkaloid extract with significantly higher antifungal activity. The minimum inhibitory activity was recorded at 1000 ppm. The antifungal activity of alkaloid extract at 2000 ppm or even at lower dose was more effective than the recommended dose of the synthetic fungicides blitox, captan, dithane M-45 and thiram [34]. The acetone, chloroform, diethyl ether, methanol, ethanol and DMSO extract of *P. juliflora* also revealed interesting antimicrobial effects. DMSO extract exhibited the higher antibacterial effects against *E. coli* (21 mm), *Serratia marcescens* (16 mm), *S. aureus* (17.9 mm), *P. fluorescens* (16.5 mm), *Paeciliomyces variotii* (13.2 mm) and *Phomopsis leptostromiformis* (11 mm). Similarly, the methanol extract exhibited high potency against *B. subtilis* (23 mm) and *Pestalotia foedans* (16 mm). Ethanol extract exhibited higher activity against *K. pneumoniae* (11 mm); however, no extract exerted activity against the fungi *Fusarium oxysporum* [35]. The antimicrobial activity of alkaloid-enriched extracts from *P. juliflora* pods was also studied for their value as feed additives for rumiants. The basic chloroformic extract [main constituents—juliprosopine, prosoflorine and juliprosine] exhibited Gram-positive antibacterial effects against *M. luteus* (minimum inhibitory concentration (MIC) = 25 μg/mL), *S. aureus* (MIC = 50 μg/mL) and *S. mutans* (MIC = 50 μg/mL), and its effect was analyzed on ruminal digestion by a semi-automated in vitro gas production technique, with monensin as positive control. Results displayed that the extract reduced gas production as efficiently as monensin 36 h post-fermentation [36].

##### Anthelmintic, Antiprotozoal and Antiplasmodial Effects 

Anthelmintic property of microencapsulated *P. juliflora* extracts was assessed against *Haemonchus contortus* extracted from naturally infected sheep. Ethanolic extracts of roots and leaves of the species were evaluated, and the results revealed that microencapsulated leaves ethanolic extract (at ratio 2:1), leaf ethanolic extract and albendazole (2.0 mg/mL dose) displayed 100% inhibition of egg hatchability. However, no differences were recorded in mean percentage egg hatch inhibition on both leaf and root ethanolic extracts as compared to albendazole. In larval mortality assay, all microencapsulated extracts of *P. juliflora* (leaves and roots) stimulated over 50% mortality of larvae at 2 mg/mL. Albendazole needed a maximum concentration of 0.25 mg/mL to produce 100% larval mortality. Further, there was a significant variation in larval mortality than egg hatchability. Adult mortality assay suggested that there was a significant difference in mean percentage adult mortality of *H. contortus* at varying doses and ratios. All assays exhibited dose-dependent response. The EC_50_ values from microencapsulated leaves and roots extracts on adult mortality assay ranged from 1.95 to a maximum of 26.87 mg/mL. These results were significantly different than that of albendazole (0.05 mg/mL) and leaves ethanol extracts (0.71 mg/mL). These findings demonstrated that microencapsulated *P. juliflora* extracts had anthelmintic activity on eggs, larvae and adults of *H. contortus* parasite [37]. The in vitro anthelmintic activity of the alkaloid containing fraction of *P. juliflora* pods was assessed on goat gastrointestinal nematodes by egg hatch, larval migration inhibition, and larval motility assay. Juliprosopine was identified as the major alkaloid in the alkaloid-rich fraction (AF) obtained from ethyl acetate extract after fractionation in chromatographic column and its characterization using nuclear magnetic resonance (NMR). Various doses were screened in egg hatch, larval migration inhibition and larval motility assay.

Cytotoxicity on Vero cell cultures was measured using 3-(4,5-dimethylthiazol-2-yl)-2,5-diphenyltetrazolium bromide and trypan blue tests. The alkaloid-rich fraction exhibited high ovicidal activity (IC_50_ = 1.1 and 1.43 mg/mL). On the contrary, low larvicidal activity and high toxicity was observed to this fraction. Therefore, it was concluded that *P. juliflora* pod alkaloid rich-fraction possess in vitro ovicidal activity against goat gastrointestinal nematodes and cytotoxicity in Vero cell cultures [37].

The in vitro antiprotozoal activity of *P. juliflora* extracted in methanol was also investigated against erythrocytic schizonts of *Plasmodium falciparum*, intracellular amastigotes of *Leishmania infantum* and *Trypanosoma cruzi* and free trypomastigotes of *T. brucei*. Cytotoxic activity was analyzed against MRC-5 cells to assess selectivity. The criterion for activity was an IC_50_ <10 μg/mL (<5 μg/mL for *T. brucei*) and a selectivity index >4. The antiplasmodial activity was detected in Amastigotes of *T. cruzi* and *T. brucei* [46].

##### Autophagy and Apoptosis Effects

*P. juliflora* leaves extracts, mostly composed of juliprosopine, were assessed for their triggering effects in programmed cell death and autophagy in a model of neuron/glial cell co-culture. It was found that total alkaloids extract (30 μg/mL) and fractions (7.5 μg/mL) led to a decrease in ATP levels and alterations in mitochondrial membrane potential at 12 h exposure. Moreover, alkaloids extract and fractions stimulated caspase-9 activation, nuclear condensation and neuronal death at 16 h exposure. The stimulation of autophagy was stated after 4 h, characterized by a reduction in P62 protein level, increase in LC3II expression and decline in the number of GFP-LC3 cells [64].

In a study, five bioactive compounds (i.e., 2-pentadecanone, butyl 2-ethylhexyl phthalate, not a natural compound methyl 10-methylheptadecanoate, methyl oleate, and phorbol-12,13-dihexanoate) from *P. juliflora* were tested against BCL2 protein using molecular docking approach. Phorbol-12,13-dihexanoate exhibited the best docking score followed by methyl oleate, showing the therapeutic importance of phytochemicals extracted from *P. juliflora* against anti-apoptotic BCL2 protein, the major target protein on cancer therapy [60].

#### 4.1.2. *Prosopis cineraria*

##### Antioxidant Effects

The antioxidant activity of different parts (stem, leaf, and bark extracts) from *P. cineraria* was assessed and the results revealed an activity similar to the standard compound ascorbic acid [65]. In another study, the ethanolic extract of *P. cineraria* were analyzed for its antioxidant potential using DPPH free radical scavenging activity [66].

##### Analgesic Effects

The analgesic properties of ethanolic extracts from *P. cineraria* roots (200 and 300 mg/kg, orally) were studied in rats using in vitro hot-plate method and tail-immersion methods. A significant analgesic effect was observed to ethanolic extracts at 200 mg/kg, being this effect comparatively more effective than the higher dose (300 mg/kg b.w.) in both assays [43].

##### Antiplasmodial Effects

*P. cineraria* crude extracts from dried leaves, stem, flowers and roots were sequentially extracted in methanol, chloroform, hexane, ethyl acetate and water using the Soxhlet method, and tested against chloroquine (CQ)-sensitive *Plasmodium falciparum* 3D7 strain and cytotoxicity against THP-1 cell line. Ethyl acetate (leaf, stem, flower and root) and chloroform extract (root) exhibited IC_50_ values of 5–50 μg/mL with good antimalarial properties. Chloroform (leaf, stem, flower) and aqueous (stem, flower and root) extracts exhibited IC_50_ values ranging from 50 to 100 μg/mL. Out of all the extracts, the ethyl acetate extract of flower (IC_50_ = 27.33 μg/mL) showed excellent antimalarial properties. The methanol and hexane extract of leaf, stem, flower and root and aqueous extract of leaves did not exhibit any activity. However, all extracts were non-toxic to THP-1 cells. Thus, the study revealed that ethyl acetate extract of flower may be important to the synthesis of antimalarial drugs [44].

##### Antimicrobial Effects

The antibacterial activities of *P. cineraria* aerial portions were determined against some human pathogenic bacteria. The ethyl acetate fraction exhibited highest antibacterial properties, and the observed activity was attributed to the presence of 2 substances with molecular weight of 348 and 184 Dalton that prevented Gram-positive bacteria growth, with MIC values <125 and <62.5 μg/mL [54]. Similarly, the pods extract of *P. cineraria* extracted in chloroform and benzene were screened against three Gram positive (*B. subtilis*, *S. aureus*, *Mycobacterium smegmatis*) and three Gram negative (*P. aeruginosa*, *K. pneumoniae*, and *E. coli*) bacteria. Chloroform pods extract was active against *K. pneumoniae*, while benzene was effective against *K. pneumoniae*, *E. coli* and *B. subtilis* [55]. On the other side, and more recently, the bioengineered silver and copper nanohybrids containing *P. cineraria* leaf extract exhibited increased antimicrobial property against Gram-positive and Gram-negative MDR human pathogens [57]. In addition, an antifungal protein (38.6 kDa) isolated from *P. cineraria* seeds displayed antifungal potential against the rotten fruit pathogen, *Lasiodiplodia theobromae*, and *Aspergillus fumigatus* [50].

##### Anticancer Effects

More recently, the bioengineered silver and copper nanohybrids containing *P. cineraria* leaf extracts revealed potent cytotoxic activity against MCF-7 cancer cell line, with IC_50_ values of 65.27, 37.02 and 197.3 μg/mL for PcAgNPs, PcCuNPs and *P. cineraria* leaf extracts, respectively [57].

#### 4.1.3. *Prosopis farcta*

##### Antioxidant Effects

In a study using different solvent extracts from *P. farcta* aerial parts revealed that, *n*-hexane, methylene chloride, ethyl acetate and *n*-butanol extracts possess 83.1, 82.0, 87.2 and 87.0% inhibition percentages (I%) using the ABTS (2,2’-azino-bis-3-ethylbenzthiazoline-6-sulphonic acid) radical assay as compared to ascorbic acid (89.2%) [40]. Similar findings were also stated for *P. farcta* fruit extract, with antioxidant effects being directly attributed to its high content in phenols and flavonoids [8]. The methanolic extract of the stem bark was analyzed and compared with ascorbic acid, and the results revealed that this extract is an easily accessible source of natural antioxidants for food supplements or pharmaceutical industry formulations [67].

##### Antimicrobial Effects

*P. farcta* (aqueous and ethanolic extract) was screened for potential antibacterial activity against methicillin-resistant *S. aureus* (MRSA), with great achievements: the minimum bactericidal concentration (MBC) of aqueous and ethanolic extracts was, respectively, 100, 125 mg/mL and 25, 112.5 mg/mL [48]. The *n*-hexane and methylene chloride extract obtained from *P. farcta* aerial portion exhibited moderate antimicrobial properties. *n*-Hexane extract exhibited activity against *Shigella* spp., *E. coli* and *P. vulgaris*, and the methylene chloride extract against *Erwinia* spp., *E. coli* and *S. epidermis*. On contrary, the ethyl acetate extract displayed higher antimicrobial properties against *Shigella* spp., *E. coli*, and *C. albicans*. Likewise, *n*-butanol extract displayed higher effects against *Shigella* spp., *Erwinia* spp., *E. coli*, *P. vulgaris*, *S. epidermis* and *C. albicans* [40].

##### Anticancer Effects

Different extracts from *P. farcta* aerial parts were evaluated for anticancer activity against four human tumor cell lines (HepG-2, HeLa, PC3 and MCF-7). The *n*-butanol extract exhibited the highest activity against the MCF-7 cell line (IC_50_ = 5.6 μg/mL) as compared to 5-fluorouracil (IC_50_ = 5.4 μg/mL). The ethyl acetate extract demonstrated the highest effect against Hela cell line (IC_50_ = 6.9 μg/mL) when compared to 5-fluorouracil (IC_50_ = 4.8 μg/mL) [40].

#### 4.1.4. *Prosopis glandulosa*

##### Antiplasmodial Effects

Two new indolizidine alkaloid, named Δ^1,6^-juliprosopine and juliprosine, were extracted from *P. glandulosa* leaves. Both compounds exhibited potent antiplasmodial activity and no toxicity was stated on VERO cells up to a dose of 23.800 ng/mL. The antileishmanial property of indolizidines was studied against *Leishmania donovani* promastigotes, axenic amastigotes and amastigotes in THP1 macrophage cultures. Tests against macrophage cultures revealed that the tertiary bases (Δ^1,6^-juliprosopine, prosopilosine, juliprosine) were more potent as compared to quaternary salts (diastereoisomer prosopilosidine, isoprosopilosidine, juliprosine), exhibiting IC_50_ values between 0.8 and 1.7 μg/mL and 3.1–6.0 μg/mL, respectively. In addition, juliprosine displayed potent antifungal activity against *Cryptococcus neoformans* and antibacterial effects against *Mycobacterium intracellulare*, while Δ^1,6^-juliprosopine revealed potent activity only against *C. neoformans*, while weak activity was detected against other organisms [45].

##### Antimicrobial Effects

*P. glandulosa* leaves ethanolic extract also displayed a good antimicrobial potential against *C. neoformans* (30.6 mm), *C. albicans* (20.0 mm), *S. epidermidis* (21.8 mm), *S. aureus* (17.4 mm), *Shigella flexneri* (19.8 mm), *P. vulgaris* (18.0 mm) and *Vibrio parahaemolyticus* (15.8 mm) [49].

#### 4.1.5. *Prosopis laevigata*

##### Antioxidant Effects

Acetone extracts and purified fractions from *P*. *laevigata* leaves were screened for in vitro antioxidant activity. Dewaxed Mezquite leaves were extracted with aqueous acetone (70%) and the polar extract was purified in Sep-Pak^®^ Cartridges and used for evaluation of their fractions. Significant variations in antioxidant activity were recorded among fractions and crude extracts using scavenging hydroxyl and DPPH radical assays [42].

##### Cardioprotective Effects

The acetone extracts and purified fractions from *P. laevigata* leaves were evaluated for in vitro cardioprotective activity. Mezquite leaves were dewaxed with petroleum ether and extracted with aqueous acetone (70%). The polar extract of species was purified in Sep-Pak^®^ Cartridges and derived fractions were analyzed. Purified fractions displayed antihypertensive activity, being able to prevent angiotensin converting enzyme and inhibited low density lipoprotein (LDL) oxidation [42].

#### 4.1.6. *Prosopis flexuosa*

##### Antioxidant Effects

The biological property of extracts from the aerial portions of five Argentinian *Prosopis* species and *P. flexuosa* exudate were analyzed for DNA binding, β-glucosidase inhibition and free radical scavenging activity using the DPPH assay. DNA binding activity was detected mainly in the basic fraction. The alkaloids tryptamine, as well as piperidine and phenethylamine derivatives were extracted from the basic extracts. At 0.50 mg/mL, DNA binding properties ranged from 28% (tryptamine) to 0–27% (phenethylamine) and 47–54% (piperidine derivatives). Tryptamine and 2-β-methyl-3-β-hydroxy-6-β-piperidinedodecanol displayed a moderate inhibition (27–32%) on β-glucosidase enzyme at 100 μg/mL. *P. flexuosa* exudate demonstrated a strong free radical scavenging activity in the DPPH assay [41].

#### 4.1.7. *Prosopis africana*

##### Antimicrobial Effects

Antimicrobial properties of the aqueous and ethanol extracts of *P. Africana* root and stem were investigated against various microorganisms, like *C. albicans*, *S. mutans* and *S. saprophyticus*. Both ethanol and aqueous extracts revealed inhibitory activity against the screened microorganisms. A significantly higher inhibitory activity of aqueous and ethanol extracts of the stem extract was recorded against *C. albicans*. Similarly, the ethanol extract showed a significantly higher inhibitory effect against *C. albicans* as compared to water extract. However, both aqueous and ethanol extracts were unable to produce significant inhibitory activity against *S. mutans* and *S. saprophyticus.* Likewise, the activity of stem and root extracts against *S. mutans* and *S. saprophyticus* did not differ [47].

##### Anti-Trypanosomal Effects

Petroleum ether, chloroform, methanol and aqueous extracts from leaves, stem bark and roots of *P. africana* were screened for anti-trypanosomal effects. Strong in vitro anti-trypanosomal activity was displayed by all solvent extracts at 2 and 4 mg/mL. Hence, *P. africana* extracts has significant anti-trypanosomal activity to warrant bioassay-guided analysis and detection of the active principle [59].

#### 4.1.8. *Prosopis alba* and *Prosopis nigra*

##### Antioxidant Effects

In a study, the sugar-free polyphenolic extracts of *P. nigra* and *P. alba* obtained from edible pods and anthocyanins enriched extracts of *P. nigra* demonstrated high antioxidant activity. Polyphenolic extracts of *P. nigra* and *P. alba* exhibited a prominent activity against a pro-inflammatory enzyme [10]. According to Cardozo and co-authors, aqueous extracts of *P. nigra* flour showed better antioxidant capacity to quench ABTS^+^ radicals (6161.93 μM Trolox/100 g) compared to *P. alba* flour (5706.10 μM Trolox/100 g), whereas alcoholic extracts of *P. nigra* exhibited lower values (236.60 μM Trolox/100 g) than *P. alba* (582.08 μM Trolox/100 g) [68]. Moreover, in a recent study, Vasile et al. (2019) also showed that *P. alba* exudate gums contained naturally-occurring phytochemicals, namely total polyphenols (9.55 mg GAE/g, flavonoids (2.53 mg QE/g, total tannins, 2.10 mg tannic acid eq./g, condensed tannins 0.61 mg quebracho tannins eq./g) with promising antioxidant potential (OH radical inhibition, 50.61 %/mg, Reducing power, 2.19 mg ascorbic acid eq./g, Fe^3+^ chelating capacity 0.14 mg EDTA eq./g gum) [11].

#### 4.1.9. *Prosopis kuntzei* and *Prosopis ruscifolia*

##### Antimicrobial Effects

*P. kuntzei* and *P. ruscifolia* extracts also revealed interesting antimicrobial effects, being able to prevent bacterial growth effectively, and to decrease the initial number of viable counts by at least one order of magnitude in 10 h [53].

### 4.2. In vivo Biological Effects of Prosopis Plants

As discussed in the previous subsection, the number of pre-clinical studies, mostly those in vivo performed is increasing, with the intent not only to confirm and support the in vitro findings, but also to trigger in-depth clinical studies. These are briefly listed in Table 4 and discussed below.

#### 4.2.1. *Prosopis cineraria*

##### Antidiabetic Effects

Two new compounds (methyl 5-tridecyloctadec-4-enoate and nonacosan-8-one), together with three known compounds (lupeol, β-sitosterol and stigmasterol) were isolated from chloroform fraction of *P. cineraria* stem bark. The chloroform fraction of stem bark was investigated in STZ-stimulated experimental diabetic rats, at doses of 50 and 100 mg/kg b.w. for 21 days. A marked reduction in blood glucose levels, glycosylated hemoglobin was observed, and it was also able to restore body weight, liver glycogen content and serum insulin level in diabetic rats, in a dose-dependent manner. Furthermore, a decrease in serum lipid profile and elevation in HDL after administration of the chloroform fraction was found, revealing that chloroform fraction has potential to protect from diabetes-associated complications [69]. *P. cineraria* bark extract antidiabetic effects were also assessed in male Swiss albino mice with alloxan-stimulated diabetes. The administration of the crude ethanolic extract of bark for 45 days remarkably reduced blood glucose levels, enhanced hepatic glycogen content and maintained body weight and lipid-profile attributes up to near normal levels. The activity of antioxidant enzymes and the amount of non-enzymatic antioxidants were also normalized, thereby decreasing the oxidative damage in tissues of diabetic animals, and hence suggesting the antidiabetic and antioxidant properties of the extract [71]. Likewise, the hypolipidemic and anti-atherosclerotic activity of *P. cineraria* bark extract was investigated in hyperlipidemic rabbits. *P. cineraria* bark extract (70% ethanol) supplementation significantly decreased serum total cholesterol (TC) (88%), low density lipoprotein cholesterol (LDL) (95%), triglyceride (59%), very low-density lipoprotein cholesterol (VLDL) (60%) levels and ischemic indices. Atherogenic alterations were markedly inhibited by *P. cineraria* bark extract in aorta. Toxicity profile parameters were also studied and were under normal range. Thus, *P. cineraria* bark has hypolipidemic and anti-atherosclerotic effects along with non-toxic nature [72]. In another study, the hypolipidemic potential of the ethanol (70%) extract of *P. cineraria* fruit was investigated in Triton-simulated hyperlipidemic rats. Whole dried fruits were pulverized to obtained extract using 70% ethanol. Adult Sprague Dawley rats were divided into six groups (*n* = 6 rats/group), named as normal control, hyperlipidemic control, standard drug-treated (simvastatin 4 mg/kg), and three ethanol extract (200, 400 and 600 mg/kg, respectively)-administrated groups. Besides normal control, all other groups received a single dose of triton (200 mg/kg, i.p.) exactly 30 min after a dose of the standard drug and ethanol extract for stimulating hyperlipidemia. Triton-stimulated hyperlipidemic group significantly enhanced TC, LDL, VLDL, triglyceride, atherogenic index and reduced high density lipoprotein cholesterol (HDL) levels than the normal control group. Ethanol extract-supplemented groups demonstrated decrease in serum TC, triglyceride, VLDL and LDL levels than the triton-administrated control group. At 200 mg/kg, the fruit extract significantly decreased serum TC and LDL levels. At higher doses (400 mg/kg and 600 mg/kg), the extract significantly decreased serum TC, triglyceride, VLDL and LDL levels and atherogenic index, being these findings almost equivalent to those of the standard drug simvastatin. Molecular docking score revealed excellent binding conformation of extract to receptor molecules [70]. These findings reveal that this plant extract has promising effects on both prevention and treatment of hyperlipidemic complications.

##### Wound Healing Effects

The wound healing properties of *P. cineraria* leaves ethanol extract on excision wounds stimulated in rats was assessed over a period of 13 days. Sulphathiazole ointment was used as a standard drug. Topical application of ethanol extract led to a 92.13 ± 3.23% decrease in wound area as compared to control (91.45 ± 5.23%), suggesting that this extract exert significant pro-healing properties by enhancing the healing process at various phases of tissue repair [66]. Similarly, in another study, the wound healing properties of *P. cineraria* ethyl acetate, chloroform and butanol fractions was assessed in rats [76]. Butanol fraction was found the most active fraction, revealing significant antioxidant and anti-inflammatory, as well as anti-collagenase and anti-elastase activities. The study concluded that the butanol fraction has ability to act as an effective cutaneous wound healing agent, promoting a faster wound repairing process, raising hydroxyproline content, decreasing the epithelialization period and the levels of inflammatory markers in blood [76].

##### Antipyretic Effects

*P. cineraria* leaves and fruits ethanol extracts (200 and 300 mg/kg b.w.) reduced brewer’s-yeast induced pyrexia in albino rats. Leaves extract also exhibited significant activity in lowering the rectal temperature in rats, being more pronounced than fruits extract (at 200 mg/kg), while leaves and fruit extract (at 300 mg/kg) significantly decreased pyrexia [78].

##### Spasmolytic, Bronchodilator and Vasodilator Effects

Crude methanolic stem bark extract of *P. cineraria* was screened for its spasmolytic, bronchodilator and vasodilator properties. In isolated rabbit jejunum preparations, the extract caused relaxation of spontaneous and K^+^ (80 mM)-stimulated contractions at tissue bath concentrations of 3–10 mg/mL, which may be due to Ca^+2^ channels blockade. This result was further supported by the shifting of the Ca^+2^ concentrations response curves to the rightward in a similar manner as of verapamil (standard). The extract also displayed non-specific relaxant activity on carbachol (1 μM)- and K^+^ (80 mM)-stimulated contractions in isolated rabbit tracheal preparations. The same effect was observed for phenylephrine (11 μM) and K^+^ (80 mM)-induced contractions in isolated rabbit aortic preparations, similarly to verapamil. These findings confirm that recorded bronchodilator and vasodilator activities were possibly mediated through Ca^+2^ channels blockade [80].

##### Anti-Depressive and CNS Disorders Effects

In a study, forced swim test (FST) was used to assess the antidepressant activity of *P. cineraria* extract, and compared with imipramine. Control and treated mice were investigated for immobility periods. Muscle relaxant activity was analyzed using rotarod apparatus and total fall off-time for standard and control group was measured. A significant decrease in immobility time duration in FST was observed (at a dose of 200 mg/kg of leaf extract), and the effect was similar to imipramine [81].

##### Skin Caring and Antiaging effects

*P. cineraria* bark extract (2% bark extract loaded emulsion formulation) was investigated using non-invasive probe cutometer and elastometer as compared to base formulation. The preliminary results indicated that bark extract possess a significant amount of phenolics and flavonoids with enormous antioxidant activity. Also, it demonstrated an efficient antibacterial, lipoxygenase, and tyrosinase enzyme prevention activities. Importantly, the bark extract did not show any toxicity or apoptosis, when incubated with HaCat cells. Moreover, in vivo results revealed that the formulation (size 3 μm) reduced skin melanin, erythema and sebum contents up to 2.1- and 2.7- and 79%, while reduced the skin hydration and elasticity up to 2-folds and 22% than the base, respectively. Owing to increased therapeutic activity, the phytocosmetic formulation proved to be a potential skin whitening, moisturizer, anti-acne, anti-wrinkle, anti-aging therapy, besides to be able to stimulate skin rejuvenation and resurfacing [87]

##### Anthelmintic Effects

The anthelmintic efficacy of *P. cineraria* bark petroleum ether, methanol and water extracts was independently evaluated on adult Indian earthworm *Pheretima posthuma*, using albendazole as standard. The time needed for earthworms’ paralysis and death was recorded. *P. cineraria* methanol extract was more effective as compared to petroleum ether and water extracts [58].

#### 4.2.2. *Prosopis glandulosa*

##### Antidiabetic Effects

*P. glandulosa* extract was also investigated for its efficiency against diabetes in male Wistar rats rendered with (1) type 1 diabetes after an STZ injection (40 mg/kg i.p) and (2) insulin resistance after a 16-week high caloric diet (DIO). Zucker fa/fa ZDF rats were used in a detailed study. Half of animals of each group were kept on *P. glandulosa* administration (100 mg/kg/day) for 8 weeks and the remaining ones were used as controls. *P. glandulosa* administration significantly (*p* < 0.001) enhanced insulin levels, supported by a remarkable reduction in blood glucose levels. Further, *P. glandulosa* administration significantly enhanced small β-cells in pancreas. A marked reduction in body weight of the STZ-treated rats after STZ injection was practically inhibited with *P. glandulosa* administration. In case of Zucker fa/fa rats, *P. glandulosa* supplementation significantly decreased fasting glucose levels and attenuated IPGTT, as compared to untreated animals. In DIO insulin resistant model, *P. glandulosa* administration enhanced basal and insulin-induced glucose uptake by cardiomyocytes prepared from this group. These result revealed that *P. glandulosa* administration moderately lowered glucose levels in different animal models of diabetes, induced insulin secretion, triggered small β-cells formation and attenuated insulin sensitivity of isolated cardiomyocytes [73].

##### Inflammation and Regeneration

The effects of oral *P. glandulosa* administration on inflammation and regeneration in skeletal muscle after contusion injury were investigated and compared to a conventional treatment. *P. glandulosa* (100 mg/kg/day) was administered in rats either for 8 weeks prior to injury (up until day 7 post-injury), only post-injury, or with topically applied diclofenac post-injury (0.57 mg/kg). Neutrophil (His48-positive) and macrophage (F4/80-positive) infiltration was measured. Muscle satellite cell proliferation (ADAM) and regeneration (desmin) indicators were used to analyze muscle repair. Chronic *P. glandulosa* and diclofenac supplementation (*p* < 0.0001) was associated with neutrophil response suppression in the contusion injury. However, only chronic *P. glandulosa* administration facilitated muscle recovery more effectively (increased ADAM and desmin expression), while diclofenac supplementation had inhibitory effects on repair, despite effective on neutrophil response inhibition. Thus, *P. glandulosa* administration results more effective in muscle repair after contusion [86].

##### Antimalarial Effects

A potent anti-infective and antiparasitic indolizidine isolated from *P. glandulosa*, called prosopilosidine, showed potent in vivo activity in a murine model of cryptococcosis (at 0.0625 mg/Kg/day/ip for 5 days) by eliminating 76% of *C. neoformans* infection from brain tissue compared to ~83% of amphotericin B (at 1.5 mg/Kg/day). Prosopilosine also revealed in vivo antimalarial activity (ED value of ~2 mg/Kg/day/ip) against *Plasmodium berghei*-infected mice 3 days post-treatment [89].

##### Antimicrobial Effects

A compound extracted from *P. glandulosa*, identified as the 2,3-dihydro-1*H*-indolizinium alkaloid prosopilosidine (PPD), was also assessed against *C. neoformans* in a murine model of cryptococcosis. Mice infected with live *C. neoformans* were administered with PPD once a day (i.p.) or twice a day (bid) orally, or with amphotericin B (Amp B) intraperitoneally (IP), or with fluconazole (Flu) orally for 5 days 24 h post-infection. Live *C. neoformans* was recorded from brains of all animals. PPD displayed potent in vivo activity against *C. neoformans* (at 0.0625 mg/kg dose) by eliminating ~76% of the organisms compared to ~83% using Amp B (1.5 mg/kg). In addition, compound from the species either 0.125 or 0.0625 mg/kg was found to be equally efficacious and less toxic as compared to Amp B (1.5 mg/kg) when it was supplemented bid (twice a day) by an i.p. route. However, PPD (10 mg/kg) exhibited potent activity when tested by an oral route with ~82% of organisms removal from the brain tissue, whereas Flu (15 mg/kg) decreased ~90% of infection [90].

#### 4.2.3. *Prosopis juliflora*

##### Antipyretic Effects

*P. juliflora* ethanolic extract was also evaluated for its potential, effectiveness and safer anti-pyretic properties in male rats. There were four groups: group 1 was administrated with water for injection (100 mL/kg); group 2 supplemented with paracetamol (150 mg/kg p.o. dissolved in water for injection); group 3 and 4 were administered *P. juliflora* ethanol extract (250 and 300 mg/kg p.o., respectively). A significant decrease in rectal temperature was recorded at 3 h. Likewise, a significant decrease in rectal temperature was recorded at 2, 3 and 4 h in comparison with the vehicle control [79].

##### Antimalarial Effects

The in vivo antimalarial activity of alkaloid-enriched extracts and pure alkaloidal constituents of *P. juliflora* was assessed in *Plasmodium berghei* NK65 infection in mice via oral supplementation. Alkaloid-enriched extracts from leaves and pods showed a significant in vivo antimalarial effect with little parasitemia inhibition (at 2 mg/kg). Julifloridine was weakly active, but juliprosopine caused a parasitemia inhibition (at 2 mg/kg) similar to that recorded for chloroquine (at 50 mg/kg) [88].

##### Antimicrobial Effects

The alkaloids, juliflorine, julifloricine and benzene insoluble-alkaloidal fraction of *P. juliflora* were also assessed for their therapeutic potential after topical application in produced superficial skin infection by rubbing freshly isolated *S. aureus* from human clinical specimen onto 9 cm^2^ shaved skin. Varying doses of juliflorine, julifloricine, benzene insoluble alkaloidal fraction and gentamicin (standard drug) were prepared in petroleum jell and applied onto infected areas. Juliflorine was effective on *S. aureus* skin infection. At 0.5, 1, and 2.5% doses, juliforine were able to heal 25, 50 and 100% of skin lesions in 2 weeks, being comparatively more effective than juliflocricine. Julifloricine was less effective when compared to juliflorine; the benzene insoluble alkaloidal mixture was more effective than juliflorine. Mixture exhibited slightly faster healing properties. Both juliflorine and the mixture were effective at 2.5% concentration, but along with toxicity. Gentamicin was observed to be more effective than the alkaloids in artificially produced skin infection [56]. The antibacterial activity of *P. juliflora* crude extract with commercially available mouthrinses on oral and periodontal organisms was also assessed. The effect of *P. juliflora* was recorded as higher than the other commercial mouthrinses against the selected microbes [52].

##### Anti-Dandruff Agent

In a study, the anti-dandruff potential of *Datura metel* and *P. juliflora* shampoo and extract were evaluated using the simple standard protocol. Study revealed that both extracts were excellent antioxidant, antibacterial and antimalassezic agents. The shampoo of the species was found effective. Thus, incorporating *Datura metel* and *P. juliflora* extracts into shampoo as an antidandruff agent will be useful in treating people affected with excess dandruff. Further studies are needed to enhance the properties of the prepared shampoo [91].

#### 4.2.4. *Prosopis farcta*

##### Neuroprotective Effects

In a study, *P. farcta* pod aqueous and ethanol extracts were evaluated for neuroprotective effects on α-motoneuron neuronal density using rat model. Male Wistar rats were divided into eight groups, namely control, compression and experimental groups. The compression and experimental groups possess highly compressed right sciatic nerve for 60 s, experimental groups (compression + aqueous extract of *P. farcta*, (intraperitoneal, Table, 2 times) and (compression + ethanol extract of *P. farcta*, (i.p., 2 times). Lumbar segments of spinal cord were sampled, processed, sectioned serially and stained using toluidine blue (pH 4.65) 4 weeks post-administration. Stereological quantitative technique was used to determine the number of α-motoneurons count. Comparative assessment of neuronal density of compression and control groups revealed significant variations. Likewise, a meaning full variation was recorded between compression group and all treatment groups. These findings revealed that *P. farcta* pod aqueous and ethanol extracts have neuroprotective effects [75].

##### Cardiovascular Disorders

The effect of *P. farcta* root aqueous extract on experimental atherosclerosis was determined in rabbits fed with high cholesterol diet–stimulated hypercholesterolemia. A significant decrease in TC, triglyceride, HDL, LDL, and VLDL levels was recorded in rabbits administered with *P*. *farcta* root. Thus, this extract has beneficial effects on cardiovascular health [82]. The effect of *P. farcta* plant extract on thoracic aorta was evaluated. Contraction triggered by phenylephrine (1 μm), followed with varying dosages of plant extract were evaluated and the effect of plant extract on rat’s aorta with and without endothelium layer was estimated. The extract demonstrated a relaxing activity on contracted aorta, with the effect being concentration-dependent [85].

The efficacy of *P. farcta* aqueous extract of roots was evaluated on high cholesterol diet–induced NAFLD in experimental rabbits’ model. Male rabbits were randomly divided into 4 groups: control (fed by standard pellet) while other groups were administrated with 2% cholesterol amounts daily. Rabbits fed with high cholesterol diet were administrated with plant extract for 30 days orally. The serum lipid levels and enzymes were significantly enhanced in the high cholesterol diet groups as compared to the normal control group. Histopathological results suggested that large lipid vacuoles were formed in hepatocytes. *P. farcta* root administration significantly attenuated rabbit lipid profile and reduced liver injury, being thus an efficacious natural drug able to significantly attenuate rabbit lipid profile and to reduce liver injury in rabbits fed a high cholesterol diet [83].

*P. farcta* beans extract was used to assess its effects against acetaminophen-stimulated hepatotoxicity. Male Wistar albino rats were divided into six groups. Before supplementation of acetaminophen (600 mg/kg), two groups were pretreated with extract (50 and 75 mg/kg), two groups were administrated with acetaminophen or extract (50 and 75 mg/kg) alone, and the control group received normal saline solution. Both doses of extract significantly improved the biochemical attributes (liver function enzymes markers, aspartate aminotransferase and alanine aminotransferase, TC, triglyceride, HDL, LDL, and VLDL) to near normal levels. Thus, *P. farcta* beans extract (50 and 75 mg/kg) demonstrated an interesting hepatoprotective activity [84]. In another study, the blood samples of 10 blue-neck male ostriches (*Struthio camelus*) fed with *P. farcta* beans were collected from days 0 and 30 to investigate HDL, LDL, triglyceride, total serum protein, albumin, globulin, TC, calcium, inorganic phosphorus, activity of aspartate aminotransferase, alanine aminotransferase, and γ-glutamyl transferase levels. From days 0 to 30, HDL, total protein, and globulins levels significantly enhanced, whereas LDL, inorganic phosphorus, and γ-GT activity were significantly reduced [92].

##### Antimicrobial Effects

*Sphingomonas paucimobilis* isolates (6Nos) were extracted from 120 hospital workers’ hands in Iraq and the antimicrobial potential of *P. farcta* pods extracts was determined. The MIC value obtained was 1000 μg/mL for methanol and ethanol extracts and 1200 μg/mL for water extract. Moreover, the inhibitory effect of extracts was investigated against bacterial plasmid: three plasmid DNA bands were exhibited when treated with 1000 μg/mL for watery extract and missing one band after administration with 800 μg/mL for both methanol and ethanol extracts. These findings were also validated SDS-PAGE by differences in protein banding pattern in samples treated watery methanol and ethanol extract [51].

#### 4.2.5. *Prosopis ruscifolia*

##### Antidiabetic Effects

*P. ruscifolia* hydroalcoholic extract from aerial parts was evaluated on alloxan-induced diabetic rats for its antidiabetic effects. Different animal groups were created, and a single dose of water, extract (100 mg/Kg), tolbutamide (100 mg/Kg, *per os p.o.*) or insulin (5 IU/kg, *i.p.*) were administered. Normoglycemic rats were also treated with extract (100 mg/kg, *p.o.*). Acute toxicity was not detected in mice. Blood glucose levels were significantly decreased in hyperglycemic rats receiving the extract (single oral dose of 100 mg/Kg) after 24 h, and even during 28 days. *P. ruscifolia* hydroalcoholic extract demonstrated low toxicity and high efficacy in reducing blood glucose levels [74].

#### 4.2.6. *Prosopis strombulifera*

##### Antinociceptive Effect

*P. strombulifera* fruits ethanol, chloroform and ethyl acetate extracts were investigated for their antinociceptive effects along with their involvement in L-arginine-nitric oxide pathway in formalin-induced pain test in mice, using aspirin and morphine as standards. Chloroform (300 mg/kg), in contrast to ethanol and ethyl acetate extracts, significantly inhibited the in vivo nociceptive response. Moreover, chloroform produced a dose-dependent inhibition of the neurogenic and inflammatory phases in the formalin test, with the effect of chloroform extract being more potent in the inflammatory phase. The antinociception caused by chloroform extract (600 mg/kg, p.o.) was significantly improved by administration of mice with L-arginine (600 mg/kg, i.p.) [61]

## 5. Adverse Effects and Toxicological Attributes

In addition to the above-mentioned beneficial biological effects conferred by prosopis plants, there are some studies focusing on their adverse effects and even toxicological attributes (Table 5).

### 5.1. Cytotoxicity

The acute and subacute oral toxicity of *P. juliflora* ethanolic extract on Wistar rats was investigated at doses ranging from 50 to 500 mg/kg for acute toxicity analysis and rats were observed for any toxic symptoms for 72 h. No toxic symptoms were recorded up to 200 mg/kg. In case of subacute toxicity analysis, the ethanolic extracts were screened at a dose of 200 mg/kg orally once daily for 30 days. Subacute toxicity analysis did not show any changes in hematological, biochemical, renal and liver function attributes in experimental animals when compared to controls [93]. Also, the acute systemic toxicity of *P. cineraria* and *P. juliflora* methanolic extracts of leaves was assessed in swiss albino mice to explore their suitable doses for pharmacological testing. Both extracts were relatively safe at doses of 100 mg/kg b.w. [94].

The cytotoxicity of a total alkaloid extract (TAE) and an alkaloid fraction (F32) extracted from *P. juliflora* leaves in rat cortical neurons and glial cells were evaluated. F32 fraction was composed of a mixture of two piperidine alkaloids, namely juliprosopine (major constituent) and juliprosine. TAE and F32 were cytotoxic to cocultures (IC_50_ = 31.07 and 7.362 μg/mL, respectively). The exposure to a subtoxic concentration of TAE or F32 (0.3–3 μg/mL) stimulated vacuolation and disruption of the astrocyte monolayer and neurite network, led to ultrastructural alterations characterized by a synthesis of double-membrane vacuoles, and mitochondrial damage, associated with alterations in β-tubulin III and glial fibrillary acidic protein expression. Microglial proliferation was also recorded in cultures exposed to TAE or F32, with increasing levels of OX-42-positive cells [62]. The traditional syrup prepared from *Prosopis* pods (known as “algarrobina” or “arrope de algarrobo) in Andean countries, generally used in confectionery and local cuisine to prepare sweets and cocktails, were assessed on human lung fibroblasts and human gastric AGS cells, without toxic effects (IC_50_ >1000 μg/mL). The main phenolic components of the syrups are *C*-glycosylflavonoids, well-known as anti-inflammatory and antioxidants [95].

### 5.2. Fertility

Fertility in male and female rats treated with mesquite pod extract was evaluated and its effects compared with those of daidzein and estradiol. Mesquite pod extract enhanced the number of days in estrus and reduced lordosis intensity during proestrus. Mesquite pod extract-treated males showed low testicular and glandular weights, as well as reduced sperm motility, viability and count. Females administered with mesquite pod extract revealed a lower number of pups as compare to control females. However, 10 to 20% of pups were dead. These findings suggest that although mesquite pod extract can disrupt female, it cannot affect male fertility [98].

The effects of mesquite pod extracts were evaluated on various aspects of behavior and reproductive physiology on male rat, and compared with estradiol (E) and two isoflavones, daidzein (DAI) and genistein (GEN). The results showed that pod extracts disrupt male sexual behavior in a similar way as to DAI and GEN, but lower than that of E. E was found to be the main disruptor of sexual behavior; however, the extracts and phytoestrogens disrupted sexual behavior in a similar way to E, 40- and 50-days post-treatment. The extracts also enhanced testicular germ cell apoptosis, reduced sperm quality, testicles weight, and testosterone level, as phytoestrogens did, although estradiol caused less effects. The number of seminiferous tubules with TUNEL-positive germ cells enhanced in extracts administrated groups in a similar way to phytoestrogens groups, and E caused the higher activity. In case of E groups, the number of TUNEL-positive cells per tubule enhanced, but no such effect was observed in mesquite- and phytoestrogens-treated administered groups. Testicular atrophy was recorded in E-administered group. These findings reveal that mesquite pod extracts caused similar effects as to those of phytoestrogens in male rat reproduction [98,99].

### 5.3. Poisoning and Toxicity

*P. juliflora* is reported to be used for feeding animals and humans. However, the consumption of *P. juliflora* as main or sole source of food caused illness in animals (known as “cara torta” disease locally). Reports are available on intoxication with this plant is characterized by neuromuscular changes and gliosis. Cattle and goats experimentally intoxicated exhibited neurotoxic damage in the central nervous system. Histologic lesions were mainly characterized by vacuolation and loss of neurons in trigeminal motor nuclei. Furthermore, mitochondrial damage in neurons and gliosis was reported in trigeminal nuclei of intoxicated cattle. Studies have reproduced the main cellular changes visualized in cara torta disease using neural cell and contributed to understanding the mechanism of action piperidine alkaloids, the main neurotoxic compound in *P. juliflora* leaves and pods [96]. One study showed that total alkaloids extract and fractions administration (at 30 μg/mL) stimulated cytotoxicity characterized by a strong cell body contraction with very thin and long processes and condensed chromatin, besides to simulate accumulation of nitrite in culture medium, thus suggesting NO production stimulation. These results reveal that total alkaloid extract and fractionated alkaloids from *P. juliflora* act directly on glial cells, stimulating activation and/or cytotoxicity, inducing NO production, and thus may have effect on neuronal damages recorded in intoxicated animals [97].

Spontaneous poisoning by *P. juliflora* in sheep has also been reported. Out of 500 sheep in a flock, four adult male sheep were affected: one died spontaneously and three others were studied, euthanized and necropsied. Neurologic analysis focused particularly on motor and sensory-cranial nerve function, complete blood counts, serum biochemistry and urinalysis. Biochemical data revealed a substantial enhancement in creatine phosphokinase levels. Clinical signs included saliva drooling, dropped jaw, tongue protrusion and food loss from the mouth. Gross and histological lesions were similar to those previously reported in cattle and goats. Sheep were more resistant to poisoning by *P. juliflora* considering that it took 21 months of pod consumption to show clinical signs. No specific administration for *P. juliflora* poisoning in ruminants is available [101].

### 5.4. Allergy

Two hundred adult PAR patients were evaluated for prevalence of mesquite allergy and to assess the efficacy of conventional allergen-specific immunotherapy (ASIT) using allergy prick skin testing (PST) against a panel of 15 different aeroallergens comprising mesquite in Egypt. Patients demonstrating a positive PST response to mesquite only were used for mesquite conventional subcutaneous ASIT. Out of 200 studied patients, 86 displayed a positive PST response to mesquite allergen. Of them, 38 showed symptoms of allergy. Remarkable attenuation in symptom and medication scores were recorded in 24/38 patients 4 months post-ASIT initiation [100].

## 6. Pre-Clinical Effectiveness: Paving the Way for Clinical Studies

In our ephemeral world, between millions of yesterdays and tomorrows, medicinal plants become one of the most essential parts of modern medicine to decline the complication of human disease [102]. Various types of plants, shrubs, and trees have been critically investigated to highlight active components in their tissues, from leaf to roots [103]. Generally, the bioactive compounds isolated from medicinal plants function like synthetic drugs, and in most cases, these substances show more powerful medical effects than commercial medications [102]. Like many other magic medicinal plants, the shrubs and trees belonging to the *Prosopis* genus show a wide range of biological activities, and according to Food and Agriculture Organization (FAO) recommendations, these species are multifunctional trees possessing active constituents for improving the health quality of humans and animals [7]. In this regard, Shah et al. [104] reported that alkaloid-rich extract from *P. juliflora* (Pju) leaves have significant anti-inflammatory and antibacterial profiles under sub-clinical condition. According to their outcomes, the prepared extract was able to down-regulate inflammatory cytokine expression [104].

Since pro-inflammatory cytokines have a critical role in the development of inflammation during catastrophic diseases [105], this type of inhibition or regulation can postpone the complications of chronic diseases. Also, in the recent study by Ramazani et al. [106], the outcomes displayed that the Pju has significant in vivo and in vitro anti-plasmodial activity. Gurushidhappa and coworkers [107] also investigated the anti-cancer activity of Pju under in vitro and in vivo conditions and reported that the methanolic extract of Pju leaves could suppress breast cancer cells by inducing apoptosis and cell cycle arrest [107]. However, these results provided a new window to carefully consider the secondary metabolites isolated from Pju for clinical therapies to develop a new class of safe and effective anti-cancer drugs. Unfortunately, there has been no clinical trial regarding the clinical effectiveness of Pju in humans, and the currently available scientific records sporadically investigated pharmacological properties of Pju and similar trees from *Prosopis* genus.

In a recent systematic review by Damasceno et al. various aspects of Pju’s effectiveness for clinical studies have reviewed, and the authors mentioned that this tree (or similar species) has antioxidant, antibacterial, antimalarial, antiviral, anti-larval, insecticidal, antitumor, anti-diabetic, and anti-emetic properties [108]. The similar profile was also reviewed by Henciya et al., who suggested that *Prosopis* spp. are traditionally important trees all over the world widely used for improving the infections associated with diseases since ancient times [109]. Interestingly, Rajesh et al. reported another aspect of the clinical effectiveness of *Prosopis* plants [110]. Based on their outcomes under in vivo phase, the aqueous extract of *P. cineraria* (14 mg/kg) has critically detoxicated the crude venom of *Naja naja*, an Indian cobra. It seems that this extract can act as a powerful antidote for helping people who have been bitten by this perilous snake [110]. Other studies reported that the concentrated emulsion of *P. cineraria* bark extract also shared potential benefit for improving facial skin properties [87]. In some parts of Asia, and especially in Pakistan, *P. cineraria* (Pci) (or queen of the desert) is renowned for its medicinal uses, and natives used various tissues of this plant for many proposes [111]. Sharma et al. reported that *P. cineraria* bark extract has significant potential to lower blood glucose level and hepatic glycogen content in alloxan-induced mice [71]. Also, this extract was able to improve serum lipid parameters and decreased oxidative stress in the tissue of diabetic mice [71]. Similarly, Soni et al. reported impressive results for *P. cineraria* anti-diabetic potential in which the oral administration of Pci in STZ-induced rats (in a concentration-dependent manner) spectacularly reduced the level of blood glucose, glycosylated hemoglobin and improved serum insulin level, and liver glycogen content in diabetic rats [69]. Studies also indicated that *P. glandulosa* (Pgl) improved insulin sensitivity STZ-induced diabetic rats [73]. It seems that various types of alkaloids, especially piperidines, as well as flavonoids and non-flavonoids compounds extracted from the stem bark of this plant [39] are responsible for its biological activities under different conditions.

In addition to the abovementioned biological activities, the hydro-alcoholic extracts of leaves and stem bark of the plant *P. cineraria* potentially displayed significant anticancer activity through the inhibition of key genes involved in the pathogenesis of cancer cell lines in vivo [112]. Similarly, Maideen et al. [113] reported that methanolic extract of this plant (dose: 2000 mg/kg) was sufficient to prevent liver tumor through suppressing glycoprotein level and modulating membrane-bound enzymes [113]. Other studies have reported that various species of *Prosopis* genus showed considerable anti-inflammatory properties. Aynawuyi and coworkers [114] reported that the methanolic stem bark extract of *P. africana* (Paf) displayed considerable analgesic and anti-inflammatory activity in carrageenan-induced inflammation in rats [114]. They suggested that in a dose-dependent manner, the studied extract ameliorated the inflammation, and therefore such profile could approve the folkloric claims about the effectiveness of this plant to decline the severity of pains among native people [114]. Similar properties have also reported for Pju, and the outcomes suggested that alkaloid-rich extract from the bark of this plant significantly reduced inflammation in vivo [115]. The wound healing properties of Paf is also approved, and the researchers indicated that the stem bark of this plant potentially could alleviate the complications associated with induced wounds in rats [116]. In another exciting research, Cattaneo et al. showed that the *P. alba* (Pal) possessed a protein with significant anti-inflammatory and antioxidant activity and considered this tree as the new source of functional foods to introduce its beneficial biological activities into clinical therapies [7]. Similarly, Vasile et al. evaluated the toxicity effects of Pal gum, and there was no side effect under in vitro condition; thus concluding that Pal gum is a safe, functional food additive with admired impacts for human health [11]. In another study [117], the clinical effectiveness of *P. nigra* (Pni) for food and pharmacological industries has extensively investigated, and the results suggested that the flour of these tree has the potential to decline the complications associated with metabolic syndromes, oxidative stress, and inflammatory responses. The researchers declared that the Pni natural products are new sources of functional foods with beneficial effects for improving the health quality of the human body [117]. More than 200 studies sporadically reported the significant results of *Prosopis* genus (especially Pci), and these studies highlighted the efficacy of these species for commercialization to derive functional products for improving the complications of daily maladies [118].

Despite all the beneficial effects of the above-discussed trees, some studies have reported that the Pju pollen antigen could cause allergenicity among patients. Al-Frayh et al. reported that during the flowering season in Saudi Arabia, the pollen of Pju affected patients with bronchial asthma. Therefore, this query is a major problem for natives and the during introducing the drug-like products isolated from Pju into medicine the allergenicity effects of these compounds should also be considered by the investigators to enhance the quality of clinical studies [119]. However, to use these trees and similar species in clinical therapies, there are some entities that the researchers should consider with their studies. First, determining the effective doses for patients is one of the critical steps they should pass. As discussed in the previous lines, some of these trees shared toxicity effects for the human body either through causing allergenicity or enhancing gastrointestinal problems. To eliminate these side effects, providing safety assays for all products extracted from Prosopis genus is an essential query to fortify the quality of end products for clinical uses. Second, without large-scale clinical trials, the researchers cannot make a final decision on the effectiveness of *Prosopis* genus for the human body, so to approve the medicinal applications of all products obtained from these species local and international clinical studies should be performed to unveil all obscure facts about the beneficial or side effects of trees belonging to this family. The current literature approved the health benefits of the extracted products from *Prosopis* genus, but for reaching valuable information in detail, further investigations should be performed to persuade international organizations to confirm the efficacy of these medicinal plants for developing novel supplementary drugs.

## 7. Food Preservative Applications of *Prosopis* Plants

Consumer demands for natural and minimally processed foods, meeting “clean label” requirements, have challenged the food industry. A wealth of preservation methods, including high pressure processing, freeze drying, and plant-derived compounds, are available for use in the food industry. The application of *Prosopis* plants-derived products in food preservation is not new, but novel application methods are now available, increasing the application feasibility.

*Prosopis* plants-derived chemical components have been isolated and show great antimicrobial potential. Indeed, leaves, roots, and pods of *Prosopis* plants have been used as a major staple food for peoples living in arid regions of America and other areas of the world. For example, pods have been used to make fermented and non-fermented beverages, syrup, and flour. The functional properties of *Prosopis* pods have been elucidated and its potential use in functional foods formulation and food supplements suggested [120,121]. The extensive and prolonged use of *Prosopis* species as a food source suggests that it is not toxic for human consumption. An important attribute of the plant is its many chemical compounds that have been shown to have antioxidant and antimicrobial effects.

In terms of food preservation, most research has focused on liquid smoke produced by condensing of wood smoke created by the sawdust pyrolysis or wood chips of *Prosopis* followed by removal of carcinogenic polyaromatic hydrocarbons [122,123]. Typically, *Prosopis* liquid smoke is commonly referred to as liquid mesquite smoke. Liquid smoke can be applied directly to or incorporated into a product. Researchers have evaluated the antimicrobial properties of liquid smoke against a wide range of foodborne pathogens, including *Listeria monocytogenes*, *Salmonella*, *Yersina enterocolitica*, *Staphylococcus aureus*, and *Escherichia coli* [124,125,126]. Molds inhibition, such as of *Aspergillus parasiticus*, *Penicillium camemberti*, and *Penicillium roqueforti*, linked to cheese deterioration has also been demonstrated [127]. Specific functional properties of liquid smoke as an antimicrobial in food preservation have been discussed [122,123,128]. Thus, understanding carbonyl compounds, and acids concentrations, and pH of a liquid smoke product is essential to ensure proper food preservation and appropriate sensory characteristics. The minimum inhibitory concentration (MIC) of commercial liquid smoke samples against *E. coli*, *Salmonella* and *S. aureus* was determined [126]. Commercial mesquite liquid smoke concentrated extracts exhibited low %MIC’s of less than 1% for each pathogen. Often, an antimicrobial compound may exhibit excellent in vitro activity, but show no or very limited activity in a food matrix.

Products, such as cold smoked sockeye salmon lack a step that disables *L. monocytogenes*, if present. Liquid smoke applied to cold-smoked sockeye salmon inoculated with *Listeria* achieved a 2-log reduction by day 14, and no growth until day 35 of cooling [125]. A liquid smoke product, derived from hickory and oak, when incorporated into chicken/pork frankfurters significantly inhibited the *L. monocytogenes* growth compared to controls without liquid smoke [129]. The population of *L. monocytogenes* in control samples was approximately 5-log greater than in those containing liquid smoke [129]. These studies suggest that liquid smoke may inactivate or inhibit the growth of *L. monocytogenes* when associated to a food system.

Mold growth in cheeses may represent spoilage and food safety hazards. Some molds produce mycotoxins which may be hazardous to human health. Control of mold growth in cheeses can be achieved by applying films, resinous coatings, or spice essential oils. An alternative to those food preservation practices is the application of liquid smoke [122]. Researchers applied liquid hickory smoke or liquid mesquite smoke to Cheddar cheese samples and then inoculated the samples with one of three different molds (*Aspergillus parasiticus*, *Penicillium camemberti*, and *Penicillium roqueforti*). Results of the study showed that mesquite liquid smoke had slightly better antifungal properties than hickory liquid smoke [127].

Aside from use as a human food source, the *Prosopis* pods are also used as supplementary food ingredients for small ruminants (sheep, goats). The broad-spectrum antimicrobial activity of *Prosopis* have been well-documented in vitro and in human food preservation. Parasitic helminth infections in small ruminates can be economically devastating to farmers [130]. The resistance of gastrointestinal nematodes to synthetic anthelmintic and the high costs associated with the use of those compounds led researchers to seek alternative methods of control [131,132]. Extracts from *P. juliflora* leaf and root bark samples were evaluated for in vitro activity against mixed samples of gastrointestinal nematodes (*Haemonchus contortus*, *Trichostrongylus* spp. and *Oesophagostomum* spp.) [133]. Ethanol extracts were able to prevent parasitic eggs from hatching likely due to the presence of an array of phytochemicals that act destabilizing membranes, enhancing cell permeability and inhibiting egg hatching and larval development [133]. This finding was supported by other researchers that showed the alkaloid-rich fraction of *P. juliflora* pods was high in juliprosopine, a major alkaloid [37]. A major concern was the potential high level of toxicity associated with the alkaloid-rich fraction [36,37]. Thus, the use of novel encapsulation techniques may enhance both *Prosopis* extracts delivery and efficacy in control of gastrointestinal nematodes.

In short, the use of liquid smoke and chemical components derived from *Prosopis* (mesquite) in food preservation seems to meet consumer demands for clean labeling and use of natural rather than synthetic chemical antimicrobials. The potential exists for the delivery of *Prosopis* extracts in forms that will improve antimicrobial activity in food systems and to treat diseases.

### 7.1. Nutritional Attributes beyond Health Promotion

Domestic and commercial food processing has typically drastic effects on the antioxidants of food [134]. For instance, thermal treatment of *P. laevigata* flours: mesocarp flour, seed flour and exocarp-seed flour increased the total phenolic contents (40%, 17%, and 58%, respectively) and free-radical scavenging capacities (35%, 15% and 80%, respectively) significantly compared to their raw flours. These increased values could be related to the formation of Maillard reaction products [121]. Díaz-Batalla et al. also carried out another study on the effect of extrusion cooking on the bioactive components of *P. laevigata* flours. They have reported that the total phenolic contents in raw and extruded seed flour were 6.68 and 6.46 mg of GAE/g, respectively. In addition, DPPH radical scavenging capacity values in raw and extruded seed flours were 9.11 and 9.32 mg of ascorbic acid equivalent/g, respectively [135]. On the other hand, while investigating the level of polyphenols present in food materials, it is also important to know the bioavailability of these compounds, including their absorption in the human body [136]. Briones-Labarca et al. (2011) showed that high hydrostatic pressure treatments (500 MPa at 2, 4, 8 and 10 min) increased the bioaccessibility of the antioxidant activity (IC_50_), compared to the untreated *P. chilensis* seed sample [38].

The nutritional and functional properties as well as genotoxicity of *Prosopis* ripe pods-obtained flour processed under different condition were also evaluated. Sucrose revealed to be the main sugar present in flours obtained from *P. alba* and *P. nigra*. Approximately 2.9% and 1.4% of soluble proteins was present in decoctions and macerations, respectively. Aqueous extractions process with heating revealed the presence of free phenolics, flavonoids and condensed tannins. None of the samples contain high phytic acid amount enough to create a nutritional problem [68]. Besides this, an exudate gum collected from a South American wild tree (*P. alba*) was used as a wall material component to increase the oxidative stability of fish oil encapsulated in alginate-chitosan beads. The inclusion of the gum in the gelation media reduced the oxidative damage during storage as compared to the free oil and alginate-chitosan beads. The gum also improved wall material properties by providing higher oil retention activity during the drying step and subsequent storage [137]. Also, an aqueous extract from *P. juliflora* litter prevented the growth of standard cultures of bacteria and fungi, including a clinical isolate of *C. albicans*. The extract found to be bactericidal against *S. aureus* and seemed to be much milder against Gram negative bacteria. *Prosopis* leaf litter extract, when mixed in nutrient media, also decreased the total numbers of soil bacteria, and fungi, the numbers of cellulolytic and of symbiotically nitrogen fixing bacteria. These results indicated that soils receiving *Prosopis* litter are particularly poor from the agricultural viewpoint [138]. In another study, the semi-quantified flavonoids from germ of three Argentinean algarroba (*P. alba*, *P. nigra* and *P. ruscifolia*) were characterized, and very similar patterns of glycosylated flavonoids were observed, which support the taxonomic parentage of the species, besides to confirm their functional identity on a molecular basis, in view of use of their seed germ flour for food purposes. Apigenin 6,8-*C*-diglycoside isomers, like isoschaftoside and schaftoside, were the most abundant compounds, accounting for 3.22–5.18 and 0.41–0.72 mg/g seed germ flour, respectively. The glycosylated derivatives of (ISO) schaftoside, occurred at a lower abundance. In addition, apigenin 6,8-*C*-di-glycosides were found as potent α-glucosidase inhibitors, suggesting that food preparations obtained from *Prosopis* spp. and seed germ flour might contribute to carbohydrates digestion modulation in humans [139]. Another study performed in Mexico [6], showed that pollen of *P. juliflora* is an important source of flavonoids, that can be considered as a natural source of antioxidant compounds, and that the effect is related to the flavonol concentration, although when a high amount of flavonols is used, the extract of mesquite pollen can have pro-oxidant effects [6]. Not less interesting, the effect of *Prosopis* spp. honey was studied on both growth and fermentative activity of *Pediococcus pentosaceus* and *Lactobacillus fermentum*. *Prosopis* spp. honey was detected as an important source of phenolic compounds, particularly flavonoids, being their average content higher to other honeys. The lactic acid bacteria used in this study also revealed varying responses to the presence of honey. *P. pentosaceus* grow at honey concentrations up to 25% (*w*/*v*), whilst *L. fermentum* demonstrated high sensitivity, being impacted both growth and fermentative property. However, as a result of lactic acid bacteria fermentative capacity, the total phenolic and flavonoid content present in 6.5% (*w*/*v*) honey solutions was enhanced, ultimately improving the antioxidant property of this system [140]. Thus, according to the literature, *Prosopis* species could be a significant source of antioxidant compounds that may have beneficial health and food preservative effects. Therefore, it could be considered as a new alternative in the formulations of food supplements or functional foods.

### 7.2. Plant Growth Inhibition

Plant growth inhibitory alkaloids were also isolated from *P. juliflora* leaves extract. The I_C50_ value (concentration required for 50% inhibition of control) for root growth of cress (*Lepidium sativum* L.) seedlings was 400 μM for 3⁗-oxojuliprosopine, 500 μM for secojuliprosopinal, and 100 μM for a (1:1) mixture of 3-oxojuliprosine and 3′-oxojuliprosine, respectively. On the contrary, the minimum concentration displaying inhibitory activity on shoot growth of cress seedlings was 10 μM for 3⁗-oxo-juliprosopine, 100 μM for secojuliprosopinal, and 1 μM for a (1:1) mixture of 3-oxojuliprosine and 3′-oxojuliprosine, respectively. Among these compounds, a (1:1) mixture of 3-oxojuliprosine and 3′-oxojuliprosine demonstrated the highest inhibitory potential on cress seedlings growth. Plant growth inhibitory alkaloids, 3⁗-oxojuliprosopine, secojuliprosopinal, and a (1:1) of 3-oxojuliprosine and 3′-oxojuliprosine were isolated from *P. juliflora* leaves extract [141].

## 8. Conclusions and Future Perspectives

Our analysis of the literature reports markedly highlights the promising beneficial health effects of *Prosopis* plants, given the advances reached with concerns to their biological activities. The multiple pre-clinical studies conducted so far clearly emphasize the use of *Prosopis* plants as a rich source of extremely useful phytochemicals, particularly phenolic compounds. In this sense, among the *Prosopis* plants studied, *P. cineraria*, *P. juliflora*, *P. farcta* and *P. glandulosa* show the greatest potential, being their biological (i.e., antioxidant, antimicrobial, analgesic, anticancer, cardioprotective, antiplasmodial) effects confirmed by in vitro and in vivo studies. Studies with *P. laevigata*, *P. flexuosa*, *P. alba*, *P. nigra*, *P. kuntzei*, *P. ruscifolia* and *P. africana* are less explored, and their health benefits (mostly antioxidant and antimicrobial effects) are confirmed only trough in vitro studies. Thus, the way is far from being completed. In addition, for these plants, clinical studies are extremely scarce to effectively support their pharmacological effects and even further guide their dietary attributes. On the other hand, and despite their traditional uses, toxicological reports available carefully advice for more in-depth studies on this matter towards to improve the overall knowledge, safety windows, quality and widespread use of plants from *Prosopis* genus. Thus, more in-depth pre-clinical studies are needed to further support and confirm the effective biological effects of *Prosopis* plants with least scientific evidence, and even in those with scarce data on food applications. Not less important to anticipate is the conception of clinical studies with prospective design as a way to effectively overcome the gaps currently found to the most promising species.

## Figures and Tables

**Figure 1 biomolecules-09-00777-f001:**
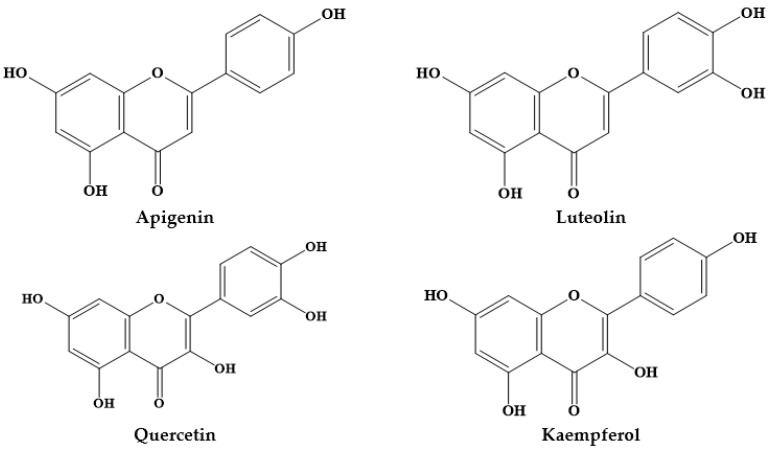
Structures of the main flavonoids in *Prosopis* plants.

**Figure 2 biomolecules-09-00777-f002:**
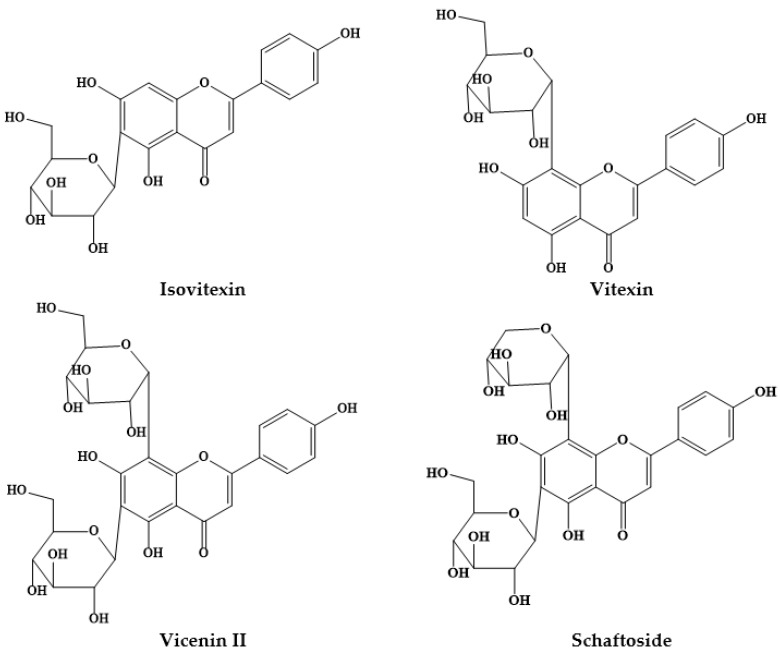
Structures of some C- glycosyl flavones from *Prosopis* plants.

**Figure 3 biomolecules-09-00777-f003:**
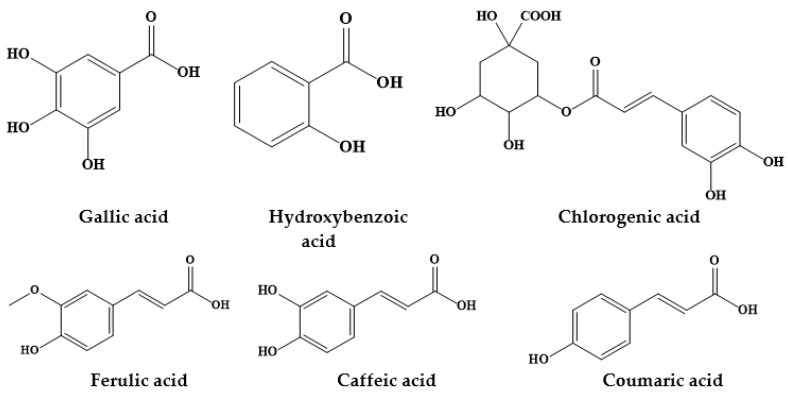
Chemical structures of some phenolic acids from *Prosopis* plants.

**Table 1 biomolecules-09-00777-t001:** Identification/quantification of phytochemicals in *Prosopis* species.

*Prosopis* Plant and Part	Identified/Quantified Phytochemicals	References
*P. alba* flours	Isovitexin (1.12–0.48 μg/mg)	[9]
	Vicenin II (1.07–0.34 μg/mg)	
	Vitexin (0.91–0.47 μg/mg)	
	Schaftoside (0.42–0.00 μg/mg)	
	Ferulic acid (4.01–0.28 μg/mg)	
	Coumaric acid (3.94–0.33 μg/mg)	
*P. alba* pods	Q-dihexoside rhamnoside	[10]
	Q-dihexoside	
	Q-methylether dihexoside	
	Vitexin	
	Q-rhamnoside hexoside	
	Isovitexin	
	Q-hexoside	
	K-hexoside	
*P. alba* flour	Isoschaftoside hexoside (2.43 mg/g)	[13]
	Schaftoside hexoside (3.33 mg/g)	
	Vicenin II/Isomer (0.67 mg/g)	
	Vicenin II/Isomer (2.34 mg/g)	
	Isoschaftoside (23.67 mg/g)	
	Schaftoside (14.86 mg/g)	
	Vitexin (0.46 mg/g)	
	Isovitexin (2.09 mg/g)	
*P. alba* exudate gum	Ferulic acid 4-glucuronide (E)	[11]
	Apigetrin, chrysin (E)	
	Chlorogenic acid (E and NE)	
	3-O-feruloylquinic acid (E)	
	p-Coumaroylquinic acid (E)	
	Valoneic acid dilactone (E)	
	Digallic acid (E)	
	Ferulic acid (NE)	
	Esculetin derivative (NE)	
	7-O-Methylapigenin (NE)	
*P. nigra* pods	Cyanidin rhamnosyl hexoside	[10]
	Cyanidin-3-hexoside	
	Peonidin-3-hexoside	
	Malvidin dihexoside	
	Cyanidin malonoyl hexoside	
	Petunidin-3-hexoside	
	Malvidin rhamnosyl hexoside	
	Malvidin-3-hexoside	
	Vicenin II	
	Q-dihexoside rhamnoside	
	Isoschaftoside	
	Q-dihexoside	
	Schaftoside	
	Q-hexoside rhamnose	
	K-hexoside rhamnoside	
	Isovitexin	
	Q-hexoside	
	K-hexoside	
	Apigenin hexoside rhamnoside	
	Q methyl ether hexoside rhamnoside	
	K-methyl ether hexoside rhamnoside	
*P. nigra* flour	Vicenin II (0.34 μg/mg)	[14]
	Schaftoside (0.24 μg/mg)	
	Isoschaftoside (0.27 μg/mg)	
	Isovitexin (0.81 μg/mg)	
	Protocatechuic acid (0.33 μg/mg)	
	Coumaric acid (8.16 μg/mg)	
	Ferulic acid (4.47 μg/mg)	
*P. cineraria*	Protocatechuic acid (31.65 mg/g) Chlorogenic acid (22.31 mg/g)	[15,16,17]
	Caffeic acid (6.02 mg/g)	
	Ferulic acid (9.24 mg/g)	
	Prosogerin A, B, C and D	
	β-sitosterol	
	Hentriacontane	
	Rutin	
	Gallic acid	
	Patulitrin	
	Luteolin	
	Spicigerin	
*P. laevigata*	Gallic acid (8–25 mg/100 g)	[18]
	Coumaric acid (335–635 mg/100 g)	
	Catechin (162.5 mg/100g)	
	Gallocatechin (340–648 mg/100 g)	
	Epicatechin gallate (10–71 mg/100 g)	
	Rutin (222.4–256.1 mg/100 g)	
	Morin (236.5 mg/100 g)	
	Naringenin (20 mg/100 g)	
	Luteolin (13 mg/100 g)	
*P. juliflora*	4′-*O*-Methylgallocatechin	[19,20]
	(+)-catechins	
	(-)-mesquitol	
	Apigenin	
	Luteolin	
	Apigenin-6,8-di-C-glycoside	
	Chrysoeriol 7-*O*-glucoside	
	Luteolin 7-*O*-glucoside	
	Kaempferol 3-*O*-methyl ether	
	Quercitin 3-*O*-methyl ether	
	Isoharmentin 3-*O*-glucoside	
	Isoharmentin 3-*O*-rutinoside	
	Quercitin 3-*O*-rutinoside	
	Quercitin 3-*O*-diglycoside	
*P. glandulosa*	Gallic acid (8.203 mg/g)	[21]
	Hydroxybenzoic acid (1.797 mg/g)	
	Pyrocatechol (5.538 mg/g)	
	Caffeic acid (0.295 mg/g)	
	Ferulic acid (0.466 mg/g)	
	Quercetin (0.045 mg/g)	

**Table 2 biomolecules-09-00777-t002:** *Prosopis* plants traditionally used in the treatment of various diseases in diverse regions of the world.

Scientific Name	Location	Local Name	Parts Used	Administration	Disease(s) Treated/Bioactive Effects	References
*P. africana*	Sélingué subdistrict, Mali	Guele	Bark trunk	Oral, Bath	Malaria	[19]
	Guinea-Bissau	Tentera, Buiengué, Bussagan, Coquengue karbon, Késeg-késeg, Paucarvão, Pócarvão, Pó-de-carbom, Po-di-carvom, Tchelem, Tchalem-ai, tchela, Tchelangadje, Tchelem, Bal-tencali, Culengô, Culim-ô, Djandjam-ô, Quéssem-quéssem, Djeiha, Ogea	Leaves, bark, roots	Unspecified	Pains, pregnancy (childbirth, breastfeeding, diseases of the newborn), skin inflammations (wounds, burns)	[20]
	Nsukka Local Government Area, South-eastern Nigeria	Ugba	Leaf	Oral	Malaria	[21]
	North-West Nigeria	Kirya, Ko-hi	Roots	Oral	Analgesic, anti-inflammatory	[22]
*P. alba*	Wichí people of Salta province, Argentina	Jwaayukw, Algarrobo blanco	Resin	Oral	Conjunctivitis, post-abortion pain	[23]
*P. cineraria*	Bahawalnagar, Punjab, Pakistan	Drucey	Leaves, stem	Oral	Spasm, diabetes, liver infection, diarrhea, removal of bladder and pancreas stone, fever, flu	[5]
				Topical	Rheumatism	
	Thar Desert (Sindh), Pakistan	Gujjo	Fruit	Oral	Tonic for body, leucorrhea	[13]
	South of Kerman, Iran	Kahour	Fruit	Topical	Asthma, skin rash	[14]
	Pakistan	Unspecified	Flower	Oral	Rheumatism	[15]
	Hafizabad district, Punjab, Pakistan	Jhand	Leaf, bark, stem, flower, fruit	Oral, topical, eye drop	Liver tonic, boils and blisters, scorpion bite, pancreatic stone, leucorrhoea, chronic dysentery, cataract	[3]
	Pakistan	Unspecified	Fruit, pods	Unspecified	Asthma	[4]
	Pakistan	Jandi, Kanda, Kandee, Jhand	Leaves, Bark, Flowers, Pods and wood	Oral	Menstrual disorders, contraceptive, prevention of abortion	[16]
*P. farcta*	Jahrom, Iran	Kourak	Fruit	Oral	Constipation, febrifuge	[24]
*P. glandulosa* Torr	Bustamante, Nuevo León, Mexico	Mezquite	Inflorescences	Oral	Stomach pain	[25]
*P. juliflora*	Thar Desert (Sindh), Pakistan	Devi	Leaves, Gum	Oral	Painkiller, boils opening, eye inflammation, body tonic, muscular pain	[13]
	Hafizabad district, Punjab, Pakistan	Mosquit pod	Whole plant, Flower, Stem, Leaves, Bark	Oral, topical, and as toothbrush	Galactagogue, kidney stones, toothache, breast cancer, asthma, boils	[3]
	Pakistan	Unspecified	Xerophytic shrub	Unspecified	Asthma, cough	[4]
	Mohmand Agency, FATA, Pakistan	Kikrye	Leaves	Oral	Lactation, expectorant	[17]
	Western Madhya Pradesh, India	Reuja	Stem bark	Oral	Asthma	[18]
*P. nigra*	Wichí people of Salta province, Argentina	Wosochukw, Algarrobo negro	Resin	Oral	Ocular trauma, conjunctivitis	[23]
*P. ruscifolia*	Wichí people of Salta province, Argentina	Atek, Vinal	Leaves	Oral	Conjunctivitis, stomachache, pimples/rash, scabies, callouses, fever, birth/postpartum pains, diarrhoea, pediculosis, otitis	[23]
*P. spicigera*	Pakistan	Unspecified	Bark, leaves, flowers	Unspecified	Asthma	[4]

**Table 3 biomolecules-09-00777-t003:** In vitro studies reporting the effect of different *Prosopis* across the globe.

Species	Model	Plant Part	Formulation/Dosage	Result	Ref.
**Antioxidant**					
*P. alba*	In vitro method	Edible pods	The sugar-free polyphenolic extracts of and obtained from edible pods and anthocyanins enriched extracts	Polyphenolic extracts of *P. alba* exhibited activity against a pro-inflammatory enzyme	[10]
*P. chilensis*	In vitro method	Seeds	Seeds were pressurized at 500 MPa during 2, 4, 8 and 10 min.	Antioxidant activity, mineral and starch content and bioaccessibility of samples were significantly affected by the processing and digestion conditions. All treatments enhanced the bioaccessibility of the antioxidant activity (IC_50_), minerals (dialysis and solubility) and starch (resistant and digestible) as compared to the untreated sample	[38]
*P. cineraria*	In vitro method	Stem bark	Methanolic extract of was analyzed and compared with ascorbic acid as reference 10.52 μg/mL (y = 0.4992x + 101.25, 0.9921) and result is 193.54 μg/mL	MPCL can be used as easily an accessible source of natural antioxidants and as a possible food supplement or in pharmaceutical industry	[39]
*P. farcta*	In vitro method	Aerial part	Oven dried material was grounded into powder (1.5 kg). Powdered materials were soaked in MeOH for 72 h followed by filtration and evaporation. Resulting crude extract was further used for solvent extraction using n-hexane, methylene chloride, ethyl acetate and n-butanol.	*P. farcta* inhibited ABTS radical in 83.1, 82.0, 87.2 and 87.0%, respectively, for the *n*-hexane, methylene chloride, ethyl acetate and *n*-butanol extracts, respectively, when compared to ascorbic acid (89.2%)	[40]
*P. flexuosa*	In vitro method		DNA binding effect was found mainly in the basic fraction. The alkaloids tryptamine as well as piperidine and phenethylamine derivatives were isolated from the basic extracts.	At 0.50 mg/mL, DNA binding activities ranged from 28% for tryptamine to 0–27% for the phenethylamine and 47–54% for the piperidine derivatives. Tryptamine and 2-β-methyl-3-β-hydroxy-6-β-piperidinedodecanol displayed moderate inhibition (27–32%) of β-glucosidase at 100 μg/mL. The exudate of *P. flexuosa* displayed a strong free radical scavenger effect in the DPPH discoloration assay, with the main active constituent identified being catechin	[41]
*P. juliflora*	In vitro method	Leaves	Ethanol extract of was investigated for antioxidant activity using in vitro DPPH assay.	Better antioxidant activity (61.55 ± 1.02 RSA %) was found as compared to the control of propyl gallate levels (88 ± 0.07 RSA %)	[26]
*P. laevigata*	In vitro method	Leaves	Leaves were extracted with aqueous acetone (70%) and the polar extract was purified in Sep-Pak^®^ Cartridges and used for evaluation of their fractions.	Significant variations were stated to antioxidant activity among fractions and crude extracts using scavenging hydroxyl and DPPH radical assays	[42]
*P. nigra*	In vitro method	Edible pods	The sugar-free polyphenolic extracts of and obtained from edible pods and anthocyanins enriched extracts	Extract demonstrated antioxidant activity. Polyphenolic extracts exhibited activity against a pro-inflammatory enzyme	[10]
**Analgesic**					
*P. cineraria*	In vitro hot-plate method and tail-immersion methods	Roots	Ethanolic extract doses (200 and 300 mg/kg, orally) were selected to study the analgesic activity.	Extracts prevented analgesic property for hot plate and tail immersion method. *P. cineraria* roots extract at 200 mg/kg was comparatively more effective at higher dose (300 mg/kg body weight) using both assays	[43]
**Cardioprotective**					
*P. laevigata*	In vitro method	Leaves	Acetone extracts and purified fractions were dewaxed with petroleum ether and extracted with aqueous acetone (70%); the polar extract was purified and their fractions analyzed	Significant variations among fractions and crude extracts were found in antioxidant capacity by scavenging hydroxyl and DPPH assays. Purified fractions displayed antihypertensive activity, preventing angiotensin converting enzyme and low-density lipoprotein oxidation	[42]
**Antiplasmodial**					
*P. cineraria*	Chloroquine (CQ)-sensitive *Plasmodium falciparum* 3D7 strain and cytotoxicity against THP-1 cell line	Leaves, stem, flowers and roots	Extraction with methanol, chloroform, hexane, ethyl acetate and aqueous sequentially. These extracts were tested in vitro against laboratory adopted *P. falciparum* 3D7 strain. The crude extracts were also tested for their cytotoxicity against THP-1 cell line.	Ethyl acetate extract of leaf, stem, flower and root and chloroform extract of root showed IC_50_ values from 5 to 50 μg/mL with good antimalarial activity. Chloroform extracts of leaf, stem, flower and the aqueous extracts of stem, flower and root showed IC_50_ values of 50–100 μg/mL. The ethyl acetate extract of flower (IC_50_ = 27.33 μg/mL) showed excellent antimalarial effects. All extracts were non-toxic to THP-1 cells	[44]
*P. glandulosa*	In vitro method	Leaves	Two new indolizidine alkaloid, named Δ1,6-juliprosopine and juliprosine were isolated. The structures of these compounds were elucidated using a combination of NMR and MS.	Compound juliprosine showed potent antiplasmodial activity against *P. falciparum* D6 and W2 strains (IC_50_ = 170 and 150 ng/mL, respectively), while Δ1,6-juliprosopine was found to be less active (IC_50_ = 560 and 600 ng/mL). Both compounds were devoid of VERO cells toxicity up to 23,800 ng/mL. When tested against macrophage cultures, the tertiary bases (Δ1,6-juliprosopine, juliprosine) were found to be more potent than quaternary salts, with IC_50_ values between 0.8 and 1.7 μg/mL and 3.1– 6.0 μg/mL, respectively	[45]
*P. juliflora*	*Plasmodium falciparum*	Leaf, bark and flower	The filter sterilized ethanol extracts at 100, 50, 25, 12.5, 6.25 and 3.125 μg/mL doses	Leaf, bark and flower extracts of *P. juliflora* showed IC_50_ values >100 μg/mL. Significant antiplasmodial activity was stated between the concentrations and time of exposure. No chemical injury was found in erythrocytes incubated with the ethanolic extract	[27]
**Antimicrobial**					
*P. juliflora*	*Plasmodium falciparum*	MRC-5 cells	Methanol extracted materials screened in vitro against erythrocytic schizonts of intracellular amastigotes of *Leishmania infantum* and *Trypanosoma cruzi* and free promastigotes of *T. brucei*. The criterion for activity was an IC_50_ <10 μg/mL (<5 μg/mL for *T. brucei*) and a selectivity index of >4.	Antiplasmodial activity was found to the extracts of *P. juliflora*	[46]
*P. juliflora*	In vitro method	Leaves	Aqueous, petroleum ether, benzene, chloroform, methanol and ethanol extracts and alkaloid extract studied using poisoned food technique against *Alternaria alternata*	Aqueous extract recorded significant antifungal activity at 24%. Among different solvent extracts screened, methanol and ethanol extract displayed significantly higher antifungal effects. Methanol extract fractionation leads to the isolation of alkaloid extract with highly significant antifungal activity against the test fungus (minimum inhibitory activity of 1000 ppm). The antifungal activity of alkaloid extract at 2000 ppm or even lower dose was highly effective than the synthetic fungicides.	[34]
*P. africana*	In vitro method	Root and stem	Aqueous and ethanol extract was investigated against various microorganisms (*C. albicans*, *S. mutans* and *S. saprophyticus*)	Ethanol and aqueous extracts of plant parts revealed inhibitory effects on the growth of microorganisms. For both aqueous and ethanol extracts, the inhibitory effect of the stem extract on *C. albicans* was significantly higher than that exhibited by the root extracts. Ethanol extract exhibited a significant higher inhibitory effect on *C. albicans* when compared to water extract. The inhibitory effects produced by the aqueous and ethanol extracts on *Streptococcus mutans* and *Staphylococcus saprophyticus* did not differ. The effects produced by the stem and root extracts on *S. mutans* and *S. saprophyticus* were not significantly different.	[47]
*P. farcta*	In vitro method	Leaf	Aqueous extract and ethanolic extract for potential antibacterial activity against MRSA isolates	MIC/IBC of aqueous and ethanolic extracts of *P. farcta* was 100, 125 mg/mL and 25, 112.5. mg/mL respectively	[48]
*P. juliflora*	In vitro method	Leaf	Silver nanoparticles (AgNPs) synthesized using an aqueous extract	Concentration-dependent activity was shown against *E. coli* and *P. aeruginosa*. Most of the studied microorganisms showed sensitivity to methanolic extract (inhibition zone ranging from 12 to 41 mm). The largest inhibition zone was detected against to *P. aeruginosa* (41 mm) and *L. monocytogenes* (33 mm) using 100 mg/mL	[28]
*P. glandulosa*	In vitro method	Leaves	Ethanolic extract against 17 microorganisms using disc diffusion method	Ethanolic extract exhibited moderate-to-high inhibitory activity against bacteria and fungi. The maximum inhibitory activity was showed against *C. neoformans* (30.6 mm), *C. albicans* (20.0 mm), *S. epidermidis* (21.8 mm), *S. aureus* (17.4 mm), *Shigella flexneri* (19.8 mm), *Proteus vulgaris* (18.0 mm) and *Vibrio parahaemolyticus* (15.8 mm)	[49]
*P. cineraria*	In vitro method		Antifungal protein (38.6 kDa) from *P. cineraria* extract was purified using ammonium sulphate precipitation, ion exchange chromatography and gel filtration	Purified protein exerted antifungal activity against post-harvest fruit fungal pathogens *Lasiodiplodia theobromae* and *Aspergillus fumigatus*	[50]
*P. farcta*	In vitro method	Aerial part	Oven dried material was grounded into powder (1.5 kg), soaked in MeOH for 72 h followed by filtration and evaporation. Resulting crude extract was further used for solvent extraction using n-hexane, methylene chloride, ethyl acetate and n-butanol.	*n*-hexane and methylene chloride extract exhibited moderate antimicrobial activities against *Shigella* spp., *E. coli* and *Proteus vulgaris* for *n*-hexane and *Erwinia spp.*, *E. coli* and *S. epidermis* for methylene chloride. Ethyl acetate displayed higher antimicrobial activities against *Shigella* spp., *E. coli*, and *C. albicans*. Likewise, *n*-butanol extract showed higher activity against *Shigella* spp., *Erwinia spp.*, *E. coli*, *P. vulgaris*, *S. epidermis* and *C. albicans*	[40]
*P. juliflora*	In vitro method		Acetone, chloroform, diethyl ether, methanol, ethanol and DMSO extract of *P. juliflora* was investigated for their antimicrobial activities. The extracts and the tetracycline as control were tested in vitro against 6 bacterial species and 4 fungal species by well diffusion method. *E. coli*, *B. subtilis*, *S. marcescens*, *K. pneumoniae*, *S. aureus*, *P. fluorescens*, *P. tromiformis var. occidentalis*, *P. foedans*, *F. oxysporum* and *P. variotii* were used	DMSO extract displayed the better antibacterial activity against *E. coli* (21 mm), *S. marcescens* (16 mm), *S. aureus* (17.9 mm), *P. fluorescens* (16.5 mm), *P. mycesvariotii* (13.2 mm) and *P. leptostromiformis* (11 mm). Methanol extract showed the better activity against *B. subtilis* (23 mm) and *P. foedans* (16 mm). Ethanol extract exhibited better activity against *K. pneumoniae* (11 mm); however, no extract displayed activity against the fungi *F. oxysporum*	[35]
*P. juliflora*	In vitro method	Pods	Antimicrobial activity of alkaloid-enriched extracts from *P. juliflora* pods	Basic chloroformic extract (main constituents were juliprosopine, prosoflorine and juliprosine) exhibited antibacterial activity against *Micrococcus luteus* (MIC = 25 μg/mL), *S. aureus* (MIC = 50 μg/mL) and *S. mutans* (MIC = 50 μg/mL). The extract reduced gas production as efficiently as monensin after 36 h fermentation, revealing positive influence on gas production during ruminal digestion	[36]
*P. farcta*	In vitro method	Pods	Six isolates of *Sphingomonas paucimobilis* were isolated from 120 hospital workers hands in Erbil city/Iraq by using VITEK2 Compact system, then further confirmed by PCR technique and by detecting specific gene TDP-glucose pyrophosphorylase (320bp) for *S. paucimobilis* ATCC 31,461 and all local isolates.	The minimum inhibitory concentration (MIC) of *P. farcta* pods extracts against *S. paucimobilis* isolate (S.p4) was 1000 μg/mL for methanol and ethanol extracts and 1200 μg/mL of watery extract	[51]
*P. juliflora*	In vitro method		Comparative assessment of antibacterial activity of crude extract of *P. juliflora* with commercially available mouthrinses on oral and periodontal organisms	*P. juliflora* activity was highest in comparison with the other commercial mouthrinses against selected microbes	[52]
*P. juliflora*	In vitro method	Seed pods	Methanol extract of *P. juliflora* at varying doses (0.05, 0.1, 0.2, 0.3, 0.4 mg mL) against *S. aureus*, *Micrococcus luteus*, *Bacillus cereus*, *Shigella sonee*, *P. aeruginosa and E. coli*	*P. aeruginosa* was the maximum resistant and *Micrococcus luteus* the less resistant to the extract	[30]
*P. juliflora*	In vitro method	Seed pods	In vitro antibacterial activity of the *P. juliflora* seed pods extract was screened against *S. aureus*, *S. epidermidis*, *E. coli* and *P. aeruginosa*	*P. juliflora* seed pods extract demonstrated antibacterial activity against all four test organisms. MIC of the extract was 0.312 mg/mL and 0.078 mg/mL, respectively for *S. aureus* and *S. epidermidis*, and 1.25 mg/mL for both *E. coli* and *P. aeruginosa*	[29]
*P. kuntzei* and *P. ruscifolia*	In vitro method		Dry extracts, dissolved in DMSO, were tested for inhibition of microbial growth via microplate assay with an oxidation-reduction dye.	*P. kuntzei* and *P. ruscifolia* exhibited MIC values ranging from to 0.08–0.5 mg dry matter/mL. All extracts at 2 × MIC were able to inhibit bacterial growth effectively, and were able to reduce the initial number of viable counts (*A. balansae*, *G. decorticans*, *P. dubium*, *G. spinosa*, *P. kuntzei* and *B. sarmientoi*) by at least one order of magnitude in 10 h	[53]
*P. cineraria*	In vitro method	Aerial parts	Organic extract was prepared via maceration in methanol, followed by the fractionation using hexane and ethyl acetate.	The best antibacterial activities were detected to the ethyl acetate fraction. The effective antibacterial constituents of the plant were two substances with molecular weight of 348 and 184 Dalton (MIC values <125 to 62.5 μg/mL)	[54]
*P. cineraria*	In vitro method	Pods	Chloroform and benzene extracts	The antimicrobial property was examined by disc diffusion assay against three gram-positive (*B. subtilis*, *S. aureus*, *M. smegmatis*) and three gram-negative (*P. aeruginosa*, *K. pneumoniae*, and *E. coli*). Chloroform pods extract was found effective against *K. pneumoniae* while benzene found effective against *K. pneumoniae*, *E. coli* and *B. subtilis*	[55]
*P. cineraria*	In vitro method		The antimicrobial alkaloids, juliflorine, julifloricine and benzene insoluble alkaloidal fraction of *P. juliflora*, were studied for their therapeutic efficacy after topical application in produced superficial skin infection. Infection was produced by rubbing freshly isolated *Staphylococcus aureus* from human clinical specimen onto 9 cm^2^ shaved skin.	Juliflorine was effective on Staphylococcal skin infection. Juliforine at 0.5, 1, and 2.5% were found to heal 25, 50 and 100% lesions in two weeks and microbiological efficacy was found to be 16.66, 33.33, 58.33 and 91.66% with 0.1, 0.5, 1 and 2.5% of juliflorine. Julifloricine was less effective when compared with juliflorine and the benzene insoluble alkaloidal mixture was comparatively more effective than juliflorine. Healing was slightly faster with the mixture. Both juliflorine and the mixture exhibited effectiveness at 2.5% concentration, however these were also found toxic. Gentamicin was found superior to the alkaloids in artificially produced skin infection	[56]
*P. juliflora*			Crude extracts were with three different solvents and examined for preliminary antibacterial activity	Varying degrees of growth inhibition were shown by all fractions. The highest antibacterial activity was observed for aqueous fractions as compared to solvent fractions.	[32]
*P. cineraria*	Human pathogens	Leaf	Characterization of silver (PcAgNPs) and copper nanoparticles (PcCuNPs)was performed using *P. cineraria* leaf extract synthesized using microwave irradiation	The bioengineered silver and copper nanohybrids showed enhanced antimicrobial activity against Gram-positive and Gram-negative MDR human pathogens.	[57]
**Anthelmintic, Antiprotozoal and Anti-trypanosomal**					
*P. juliflora*	*Haemonchus contortus* isolated from naturally infected sheep	Roots and leaves	Roots and leaves of *P. juliflora* were extracted with ethanol and evaluated for anthelmintic activity according to standard procedures.	In larval mortality assay, all microencapsulated extracts of *P. juliflora* (leaves and roots) induced over 50% mortality at the highest concentration used (2 mg/mL). Albendazole required a maximum concentration of 0.25 mg/mL to induce 100% larval mortality. There was a significant difference in larval mortality compared to that of egg hatchability. There was a marked difference in mean percentage of adult mortality of *H. contortus* at different concentrations and ratios. All assays showed dose-dependent response.	[37]
*P. juliflora*	Goat gastrointestinal nematodes	pods	In vitro anthelmintic activity of the alkaloid containing fraction	High ovicidal activity was recorded with IC and IC values at 1.1 and 1.43 mg/mL for alkaloid rich fraction. This fraction also exhibited low larvicidal activity and high toxic effect	[37]
*P. cineraria*	*Pheretimaposthuma*	Bark	Extract in methanol is prepared and used for screening	Time required for the paralysis and death was recorded. Methanol extract was more potent than petroleum ether and aqueous extracts	[58]
*P. africana*	In vitro method	Leaves, stem bark and roots	Petroleum ether, chloroform, methanol and aqueous extracts, obtained by cold extraction	All solvent extracts showed strong in vitro anti-trypanosomal activity at both 2 and 4 mg/mL	[59]
**Apoptosis**					
*P. juliflora*	BCL2 protein using molecular docking approach		Five bioactive compounds, namely 2-pentadecanone; butyl 2-ethylhexyl phthalate; methyl 10-methylheptadecanoate; methyl oleate; and phorbol-12, 13- Dihexanoate were identified using GC-MS analysis	Phorbol-12,13-dihexanoate showed best docking score of −15.644 Kcal/mol, followed by methyl oleate (−13.191 Kcal/mol)	[60]
**Antinociceptive**					
*P. strombulifera*	In vitro J774A.1 macrophage-derived cell line	Fruits	Fruit extract at varying concentrations in different solvent system	Chloroform (100 μg/mL) produced inhibition of LPS-induced NO production, which was not observed with ethanol and ethyl acetate at the same concentration. NO production inhibition by chloroform (10–100 μg/mL) was dose-dependent, (IC_50_ = 39.8 (34.4–46.1) μg/mL, and chloroform significantly inhibited LPS-induced iNOS expression in J774A.1 cells	[61]
**Anticancer**					
*P. farcta*	Cell lines namely; HepG-2, HeLa, PC3 and MCF-7.	Aerial part	Oven dried material was grounded into powder (1.5 kg), soaked in MeOH for 72 h followed by filtration and evaporation. Resulting crude extract was further used for solvent extraction using n-hexane, methylene chloride, ethyl acetate and n-butanol.	*n*-butanol extract showed the highest activity against MCF-7 cell line (IC_50_ = 5.6 μg/mL) compared to 5-fluorouracil (IC_50_ = 5.4 μg/mL), while ethyl acetate showed the highest activity against Hela cell line (IC_50_ = 6.9 μg/mL) compared to 5-fluorouracil (IC_50_ = 4.8 μg/mL)	[40]
*P. cineraria*	Breast cancer cells (MCF-7)	Leaf	Characterization of silver (PcAgNPs) and copper nanoparticles (PcCuNPs) was performed using *P. cineraria* leaf extract synthesized using microwave irradiation	MTT assay results indicated that CuNPs show potential cytotoxic effect followed by AgNPs against MCF-7 cancer cell line. IC_50_ values were 65.27, 37.02 and 197.3 for PcAgNPs, PcCuNPs and *P. cineraria* leaf extracts, respectively	[57]
**Toxicity**					
*P. juliflora*	Neurons and glial cells	leaves	Total alkaloid extract (TAE) and one alkaloid fraction (F32) at concentrations between 0.3 and 45 μg/mL were tested for 24 h on neuron/glial cell primary cocultures.	3-(4,5-dimethylthiazol-2-yl)-2,5-diphenyltetrazolium bromide test revealed that TAE and F32 were cytotoxic to cocultures (IC_50_ were 31.07 and 7.362 μg/mL, respectively). Exposure to a subtoxic concentration of TAE or F32 (0.3–3 μg/mL) induced vacuolation and disruption of the astrocyte monolayer and neurite network, ultrastructural changes, characterized by formation of double-membrane vacuoles, and mitochondrial damage, associated with changes in β-tubulin III and glial fibrillary acidic protein expression. Microglial proliferation was also observed in cultures exposed to TAE or F32, with increasing levels of OX-42-positive cells	[62]
*P. juliflora*	*Spodopteralitura larvae*	Seed pod		A significant increase in the total hemocyte count was found. *P. juliflora* seed pod hexane extract was effective in producing lepidopteran larval mortality may be due to the presence of 9-Octadecyne	[63]
*P. juliflora*	Neuron/glial cell co-culture	Leaves	A total extract (TAE) of alkaloids and fraction (F32) composed mainly of juliprosopine	TAE (30 μg/mL) and F32 (7.5 μg/mL) reduced ATP levels and led to changes in mitochondrial membrane potential at 12 h exposure. TAE and F32 induced caspase-9 activation, nuclear condensation and neuronal death at 16 h exposure. After 4 h, they induced autophagy characterized by decreases of P62 protein level, increase LC3II expression and increase GFP-LC3 cells number	[64]

ABTS, 2,2’-azino-bis-3-ethylbenzthiazoline-6-sulphonic acid; DMSO, dymethylsulfoxide; DPPH, 1,1-diphenyl-2-picrylhydrazyl; IBC, inhibitory bactericidal concentration; IC50, inhibitory concentration providing 50% of inhibition; LPS, lypopolyssacharide; MIC, minimum inhibitory contentration; RSA, radical scavenging activity.

**Table 4 biomolecules-09-00777-t004:** In vivo studies reporting the effect of different *Prosopis* across the globe.

Species	Model	Plant Part	Formulation/Dosage	Result	Ref.
**Antidiabetic**					
*P. cineraria*	STZ-induced diabetic rats	Stem bark	Chloroform fraction of species was orally administrated to STZ-induced diabetic rats at the doses of 50 and 100 mg/kg b.w for 21 days	Significant reduction in blood glucose, glycosylated hemoglobin levels and body weight, liver glycogen content and serum insulin level restoration, in a concentration-dependent manner. Decrease in serum lipid profile markers and elevation in HDL after administration, also evidencing protective effects in diabetes-associated complications	[69]
*P. cineraria*	Hyperlipidemic rats	Fruits	Extract was obtained by pulverizing whole dried fruits and extracting with 70% ethanol.	Decrease in serum cholesterol, triglyceride, VLDL and LDL levels. At 400 and 600 mg/kg, the extract significantly reduced serum cholesterol, triglyceride, VLDL, LDL and atherogenic index and these results are almost equivalent to those of drug simvastatin	[70]
*P. cineraria*	Male Swiss albino mice	Bark	Dried material was powdered followed by Soxhlet extraction with 50% aqueous ethanol and concentrated. Oven dried material was suspended in 20% tween 20 in normal saline for further experimentation.	Reduced blood glucose level, enhanced hepatic glycogen content and maintained body weight and lipid-profile attributes towards near normal range. Reduced antioxidant enzymes activity and concentration of non-enzymatic antioxidants, thereby decreasing the oxidative damage in the tissues of diabetic animals and hence indicating the anti-diabetic and antioxidant efficacy of the extract	[71]
*P. cineraria*	Hyperlipidemic rabbits	Bark	Rabbits were orally supplemented with high fat diet and cholesterol powder (500 mg/Kg body weight per day in 5 mL of coconut oil orally for 15 days) to create a hypolipidemic model.	Decreased serum total cholesterol, LDL, triglyceride, VLDL and also ischemic indices (TC/LDL and LDL/HDL). The Prevented the atherogenic changes in aorta. Toxicity profile parameters remained under normal ranges	[72]
*P. glandulosa*	Male Wistar rats (type 1 diabetic model)		Half of each group of animals was placed on treatment (100 mg/kg/day) for 8 weeks and the remaining animals served as age-matched controls.	Enhanced insulin levels, with a significant decrease in blood glucose levels. Increased small β-cells level in pancreas. Reduced fasting glucose levels and improved IPGTT. Increased and insulin-stimulated glucose uptake by cardiomyocytes	[73]
*P. ruscifolia*	Alloxan-induced diabetic rats	Aerial part	Different animal groups were administered with a single dose of water, extract (100 mg/Kg), tolbutamide (100 mg/Kg, *p.o.*) or insulin (5 IU/kg, *i.p.*). Normoglycemic rats were also treated with hydroalcoholic extract (100 mg/kg, *p.o.*).	No evidence of acute toxicity. Blood glucose levels were significantly (*p* < 0.01) decreased with a single oral dose (100 mg/Kg) after 24 h. Blood glucose levels decreased significantly with administration of plant extract during 28 days	[74]
**Neuroprotective**					
*P. farcta*	Male Wistar rats	Pod	Aqueous extract of *P. farcta* injections (25, 50, 75 mg/kg, ip, 2 time) and (compression + ethanol extract of *P. farcta* injections (25, 50, 75 mg/kg, i.p., 2 time) (*N* = 8).	Comparative assessment of neuronal density of compression and control groups exhibited marked variations. A meaning full variation was recorded between compression and all treatment groups	[75]
**Wound healing**					
*P. cineraria*	Rats using excision and incision wound model		Ethyl acetate, chloroform and butanol fractions of species hydroethanolic extract were assessed for their antioxidant activity using in vitro method	Butanol fraction found most active fraction against free radicals among all. Butanol fractions possess significant anti-inflammatory, anti-collagenase and anti-elastase activities. Application of butanol fraction ointment for 16 consecutive days on the dorsal wound area of rats confirmed the faster wound repairing process, higher hydroxyproline content, reduction in epithelialization period and inflammatory markers in blood as compared to control group	[76]
*P. cineraria*	Male albino wistar rats	Leaves	Wound excised rats administered with ethanolic extract for 13 days period.	Decrease in wound area as compared to control	[77]
**Antipyretic**					
*P. cineraria*	Brewer’s-yeast induced pyrexia in albino rat	Leaves and fruits	At a dose of 200 and 300 mg/kg of body weight was investigated.	Reduced hyperpyrexia to a significant level as compared to standard control. Lowered the rectal temperature of rats than fruits extract at 200 mg/kg while at dose of 300 mg/kg both leaves and fruit extract significantly decrease pyrexia.	[78]
*P. juliflora*	Male rats		Twenty-four male rats were randomly allotted to four groups (6 animals per group) and food was deprived off for 48 h water provided but before 24 h of experiment, water also withheld. Group 1 was treated with water for injection (100 mL/kg). Group 2 treated with Paracetamol (150 mg/kg p.o dissolved in water for injection). Group 3 and 4 were treated with ethanol extract of *P. juliflora* (250 and 300 mg/kg p.o respectively). Temperature maintained at ± 3 °C, for 0 to 4 h of interval at the dose of 250 mg/kg	Decreased the rectal temperature at 3 h and at dose 500 mg/kg. Reduced the rectal temperature at 2, 3 and 4 h in comparison with vehicle control.	[79]
**Spasmolytic, bronchodilator, and vasodilator activities**					
*P. cineraria*	In vivo method	Stem bark	The extract at 3–10 mg/mL doses	The extract caused relaxation of the spontaneous as well as K+ (80 mM)-stimulated contractions at tissue bath concentrations of 3–10 mg/mL in isolated rabbit jejunum preparations. Extract displayed nonspecific relaxant effect on carbachol (1 μM)- and K+ (80 mM)-induced contractions in isolated rabbit tracheal preparations.	[80]
**Depression and CNS disorder**					
*P. cineraria*	Mice	Leaf	Antidepressant effect was evaluated using Forced swim test (FST). The immobility periods of control and treated mice were recorded.	Leaf extract (200 mg/kg) significantly decreased the duration of immobility time in FST. The efficacy of tested extract was comparable to that of imipramine	[81]
**Cardiovascular disorder**					
*P. farcta*	Rabbit	Root	The study evaluated the efficacy of aqueous extract of *P. farcta* root on experimental atherosclerosis development in rabbits with high cholesterol diet–induced hypercholesterolemia.	Serum lipid parameters were significantly increased in the high cholesterol diet groups in comparison with the normal control group. Treatment with *P. farcta* root decreased total cholesterol, triglyceride, high-density lipoprotein, low-density lipoprotein, and very low-density lipoprotein levels compared to high cholesterol diet rabbits	[82]
*P. farcta*	Rabbits	Root	High cholesterol diet–induced NAFLD in rabbits as experimental model. Male rabbits randomly divided into 4 groups namely, control (fed by standard pellet) and other groups were received 2% cholesterol amounts daily. Rabbits were fed with high cholesterol diet till the serum cholesterol level reached 1800 mg/dl, then, they were treated daily with distilled water, and 0.6 mg/kg Simvastatin, or 500 mg/kg/day *P. farcta* root extracts orally by gavage for 30 days.	Serum lipid parameters and enzymes were significantly enhanced in the high cholesterol diet groups in comparison with the normal control group. Histopathological findings revealed that large lipid vacuoles were formed in hepatocytes. Treatment with *P. farcta* root significantly improved rabbit lipid profile and reduced liver injury.	[83]
*P. farcta*	Wistar albino rats	Beans	Thirty-six male Wistar albino rats weighing 220 ± 30 g were distributed into six groups. Two groups were pretreated with extract (50 and 75 mg/kg) for 7 days before administration of acetaminophen (600 mg/kg). Two were given acetaminophen or extract (50 and 75 mg/kg) alone, and the control received normal saline.	Extract at both doses significantly attenuated total cholesterol, triglyceride, high-density lipoprotein, low-density lipoprotein, and very low-density lipoprotein levels to near normal	[84]
*P. farcta*	Rat’s thoracic aorta	In vivo method	*P. farcta* plant extract was firstly prepared. Then 2 cm of rat’s thoracic aorta was dissected and was divided into 4 pieces of 5 mm. After contraction of these pieces by phenylephrine (1 μm), different dosages of plant extraction (0.5, 1 and 2 mg/mL) were examined and the effect of plant extract on rat’s aorta with and without endothelium layer was measured. Different dosages of *P. farcta* extract (1 and 2 mg/mL) at the presence and absence of *L- NAME* (a nitric oxide synthetase inhibitor) was examined.	*P. farcta* extract showed a dose-dependent relaxing effect on contracted aorta. The relaxing effect of plant extract on aorta with endothelium was more significant than that on aorta without endothelium in the different dosages. The relaxing effect of plant extract in the presence of *L- NAME* was decreased significantly. The relaxing effect of plant extract was more than that by acetylcholine.	[85]
**Inflammation and regeneration**					
*P. glandulosa*	Gastrocnemius muscle of rats		The gastrocnemius muscle of rats was subjected to mass-drop injury and muscle samples collected after 1-, 3 h, 1- and 7 days post-injury. Rats were treated with *P. glandulosa* (100 mg/kg/day) either for 8 weeks prior to injury (up until day 7 post-injury), only post-injury, or with topically applied diclofenac post-injury (0.57 mg/kg).	Chronic *P. glandulosa* and diclofenac treatment was associated with neutrophil response suppression to contusion injury, however only chronic *P. glandulosa* treatment facilitated more effective muscle recovery, while diclofenac treatment had inhibitory effects on repair, despite effective inhibition of neutrophil response.	[86]
**Skin caring and antiaging**					
*P. cineraria*		Stem, leaves, and bark	Performance of 2% bark extract loaded emulsion formulation was determined by using non-invasive probe cutometer and elastometer with comparison to base formulation	Bark extract did not induce any toxicity or apoptosis, when incubated with HaCat cells. Moreover, the formulation (size 3 μm) decreased the skin melanin, erythema and sebum contents up to 2.1-,2.7-and 79%, while increased the skin hydration and elasticity up to 2-folds and 22% as compared to the base, respectively. Owing to enhanced therapeutic effects the phytocosmetic formulation proved to be a potential skin whitening, moisturizer, anti-acne, anti-wrinkle, anti-aging therapy and could actively induce skin rejuvenation and resurfacing	[87]
**Antimalarial**					
*P. juliflora*	Murine model	Leaves and pods	Alkaloid-enriched extracts from (BCE) of *P. juliflora*, as well as FACB pure constituents (as formate salts), were obtained and assayed against *Plasmodium berghei* NK65 infection in mice via oral supplementation.	Alkaloid-enriched extracts from leaves and pods showed remarkable antimalarial activity with little parasitemia inhibition at the 2 mg/kg dose. Julifloridine was weakly active, but juliprosopine caused a parasitemia inhibition at the 2 mg/kg dose similar to that recorded for chloroquine at 50 mg/kg	[88]
**Anti-trypanosomal properties**					
*P. africana*	In vivo method	Leaves, stem bark and roots	petroleum ether, chloroform, methanol and aqueous extracts, obtained by cold extraction from the	Only the methanolic extract of leaves displayed promising anti-trypanosomal effect at 200 mg/kg dose	[59]
**Antinociceptive**					
*P. strombulifera*	Formalin-induced pain test in mice	Fruits	Fruit extract at varying concentrations in different solvent system	Chloroform (300 mg/kg), in contrast to ethanol and ethyl acetate extract, caused significant inhibition of the in vivo nociceptive response. Moreover, chloroform (100–1000 mg/kg, p.o.) produced a dose-dependent inhibition of neurogenic and inflammatory phases of the formalin test with inhibition values (at 600 mg/kg) of 42 ± 7 and 62 ± 7%, respectively. Antinociception was significantly attenuated by i.p. treatment of mice with l-arginine (600 mg/kg)	[61]

IPGTT, Intraperitoneal glucose tolerance test; LDL, low density lipoprotein; NAFLD, Non-alcoholic fatty liver disease; STZ, Streptozotocin; VLDL, very low-density lipoprotein.

**Table 5 biomolecules-09-00777-t005:** Adverse effects and toxicological attributes of *Prosopis* plants.

Species	Model	Plant Part	Formulation/Dosage	Result	Ref.
**Cytotoxicity**					
*P. juliflora*	Wistar rats		In an acute toxicity study *P. juliflora* extract was administered orally at doses ranging from 50 to 500 mg kg^−1^ and the animals were observed for any toxic symptoms for 72 h. In a subacute toxicity study ethanolic extracts of *P. juliflora* were tested at a dose of 200 mg kg^−1^ orally once daily for 30 days.	No changes in hematological, biochemical, renal and liver function parameters were stated in experimental animals of in this sub-chronic study when compared to control animals	[93]
*P. juliflora*	Cattle and goats	Leaves	Cattle and goats experimentally intoxicated presents neurotoxic damage in the central nervous system	Histologic lesions were mainly characterized by vacuolation and loss of neurons in trigeminal motor nuclei. Mitochondrial damage in neurons and gliosis was reported in trigeminal nuclei of intoxicated cattle	[96]
*P. cineraria*	Swiss albino mice	Leaves	Different doses of extract were used for screening.	The extracts of investigated plants are relatively safe at the dose of 100 mg/kg b.w.	[94]
*P. juliflora*	Astrocyte primary cultures derived from the cortex of newborn Wistar rats	In vivo method	TAE and seven alkaloidal fractions, at concentrations ranging 0.03–30 μg/ml	TAE and fractions F29/30, F31/33, F32 and F34/35 were cytotoxic to astrocytes, with EC_50_ values for the most toxic compounds, TAE, F31/33 and F32, being respectively, 2.87 2.82 and 3.01 μg/mL. Astrocytes exposed to 3 μg/mL TAE, F29/30 or F31/33 developed compact cell body with many processes overexpressing GFAP. Treatment with 30 μg/mL TAE and fractions, induced cytotoxicity characterized by a strong cell body contraction, very thin and long processes and condensed chromatin. Also, the proportion of OX-42 positive cells was increased in cultures treated with 30 μg/mL TAE or F29/30, F31/33, F32 and F34/35, with values raging from 7.27 to 28.74%. Moreover, incubation with 3 μg/mL F32, 30 μg/mL TAE, F29/30, F31/33 or F34/35 induced accumulation of nitrite in culture medium indicating induction of NO production	[97]
*P. glandulosa*	Mice		2,3-Dihydro-1*H*-indolizinium alkaloid-prosopilosidine (PPD) was studied against *C. neoformans* in a murine model of cryptococcosis. Mice were infected via the tail vein with live *C. neoformans*. Twenty-four hours post-infection, the mice were administered with PPD once a day (i.p.) or twice a day (bid) orally, or with amphotericin B (Amp B) intraperitoneally (IP), or with fluconazole (Flu) orally for 5 days	PPD showed potent in vivo activity against C. neoformans at 0.0625 mg/kg by eliminating ~76% of the organisms compared to ~83% with Amp B (1.5 mg/kg). In addition, PPD was equally efficacious, but less toxic, at either 0.125 or 0.0625 mg/kg compared to Amp B (1.5 mg/kg) when it was administered bid (twice a day) i.p. When tested by an oral route, PPD (10 mg/kg) showed potent activity in this murine model of cryptococcosis with ~82% of organisms eliminated from the brain tissue, whereas Flu (15 mg/kg) reduced ~90% of the infection.	[90]
**Fertility**					
Mesquite	Female and male rats	Pod	Female and male rats’ group as vehicle, mesquite pod extract, DAI and E2 were administered subcutaneously for 30 days.	These extracts disrupted both female and male sexual behavior in a similar way to DAI, but less than E2. Mesquite pod extract increased the number of days in estrus and decreased lordosis intensity during proestrus. Mesquite pod extract-treated males showed lower testicular and glandular weights, as well as decreased sperm motility, viability and count. In females treated with mesquite pod extract, the number of pups was lower than in control females, and 10 to 20% of pups were dead. These effects were similar to those with DAI-treatment. Despite the lower sperm quality, the fertility of mesquite pod extract- and DAI-treated males seem not to be disrupted, as they could impregnate control females	[98]
Mesquite	Male rat	Pod	The following treatments were given to groups of intact male rats: vehicle; mesquite pod extract; E; DAI; GEN.	Mesquite pod extracts disrupt male sexual behavior in a similar way to DAI and GEN, but less than E. The main disruptor of sexual behavior was E, however after 40 and 50 days of administration, extracts and phytoestrogens disrupted sexual behavior in a similar way to E. The extracts also increased testicular germ cell apoptosis, decreased sperm quality, testicular weight, and testosterone levels, as phytoestrogens did, although these effects were less than those caused by estradiol. Number of seminiferous tubules increased in extracts-treated groups in a similar way to phytoestrogens groups, and E caused the greatest effect. Testicular atrophy was only observed in estradiol-treated males. Testosterone declined in males of all experimental groups compared with control. Mesquite pod extracts cause effects similar to those of phytoestrogens in male rat reproduction, these effects were lower than those caused by E	[99]
**Allergy**					
Mesquite	PAR patients	Pollen	Patients demonstrating a positive PST response to mesquite only were used for mesquite conventional subcutaneous ASIT	86/200 patients displayed a positive PST response to mesquite allergen, of them, 38 were positive to mesquite allergen only. Remarkable attenuation in symptom and medication scores were recorded in 24/38 patients 4 months post-ASIT initiation	[100]

ASIT, Allergen-specific immunotherapy; PST, prick skin testing.

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
