# Peer review of "Prosopis Plant Chemical Composition and Pharmacological Attributes: Targeting Clinical Studies from Preclinical Evidence"

_biomolecules, 2019, doi:10.3390/biom9120777_

Round 1

Reviewer 1 Report

The Review by Sharifi-Rad et al. is a comprehensive and well structured collaborative review about chemical composition and potential pharmacological effects of compounds from Prosopis spp. plants.

In my opinion it is an interesting work, collecting and critically surveying practically almost all the studies carried out on phytochemical aspects of  Prosopis plants, which include species with a largely underestimated potential. Although very long, readers could easily extrapolate the information they need most.

This Reviewer recommends the publication of the review, provided that text is carefully revised and corrected preferentially with the aid of an English native proofreader.

Author Response

The Review by Sharifi-Rad et al. is a comprehensive and well structured collaborative review about chemical composition and potential pharmacological effects of compounds from Prosopis spp. plants.

Answer: Thank you for the overall appreciation of our work.

In my opinion it is an interesting work, collecting and critically surveying practically almost all the studies carried out on phytochemical aspects of  Prosopis plants, which include species with a largely underestimated potential. Although very long, readers could easily extrapolate the information they need most.

Answer: Thank you for the overall appreciation of our work.

This Reviewer recommends the publication of the review, provided that text is carefully revised and corrected preferentially with the aid of an English native proofreader.

Answer: The English language was carefully revised.

Reviewer 2 Report

The manuscript requires major revision.

In abstract: Nambers of Prosopis  plants, including P. africana, P. alba, P. cineraria, P. farcta, P. glandulosa, P. juliflora, P. nigra, P. ruscifolia  and P. spicigera are used in traditional medicine for the treatment of various diseases, such as asthma, birth/postpartum pains, callouses, conjunctivitis, diabetes, diarrhoea, expectorant, fever, flu, lactation, liver infection, malaria, otitis, pains, pediculosis, rheumatism, scabies, skin  inflammations, spasm, stomach ache, bladder and pancreas stone removal and so many others.....>Check the English.

Authors list out various members of the species and it has its own beneficial and non beneficial activities.

The authors needs to group individual species and to discuss its effects from invitro to preclinical.

Further, some species has lethal effects and it cannot be grouped in the same beneficial activities of other species.

So I strongly suggest to rewrite the manuscript with individual species.

Author Response

The manuscript requires major revision.

Answer: All comments raised by the reviewer were carefully addressed.

In abstract: Nambers of Prosopis  plants, including P. africana, P. alba, P. cineraria, P. farcta, P. glandulosa, P. juliflora, P. nigra, P. ruscifolia  and P. spicigera are used in traditional medicine for the treatment of various diseases, such as asthma, birth/postpartum pains, callouses, conjunctivitis, diabetes, diarrhoea, expectorant, fever, flu, lactation, liver infection, malaria, otitis, pains, pediculosis, rheumatism, scabies, skin  inflammations, spasm, stomach ache, bladder and pancreas stone removal and so many others.....>Check the English.

Answer: This sentence was rewritten.

Authors list out various members of the species and it has its own beneficial and non beneficial activities. The authors needs to group individual species and to discuss its effects from invitro to preclinical.

Answer: the revised version of the manuscript was organized according to the reviewer suggestion.

Further, some species has lethal effects and it cannot be grouped in the same beneficial activities of other species.

Answer: Toxicological data was included in a separate table and respective subsection

So I strongly suggest to rewrite the manuscript with individual species.

Answer: The revised version of the manuscript was changed accordingly.

Round 2

Reviewer 2 Report

Need Again Major Revision

From Line 784 to till conclusion.

As author desribes it has toxicity effect and how they can generalize the conclusion in the use of food?

So I strongly suggest not to focus on the food usage.

In conclusion and future perspectives.

Authors need to specify group of species for its health benefits and other effects.

Future perspectives need more from author point of view

Author Response

Need Again Major Revision

From Line 784 to till conclusion.

As author describes it has toxicity effect and how they can generalize the conclusion in the use of food? So, I strongly suggest not to focus on the food usage.

Answer: Thanks for the reviewer's suggestion. We understand your position, but as this is a comprehensive review intending to provide useful information for readers about Prosopis plants, we have attempted to summarize all available information about these plants. In fact, Prosopis plants food applications are the least explored and given this gap, toxicity can be discussed. However, we believe this information will be useful in triggering the design of more in-depth studies addressing the current gaps found towards their resolution and scientific progress.

In conclusion and future perspectives.

Authors need to specify group of species for its health benefits and other effects. Future perspectives need more from author point of view

Answer: Conclusion and future perspectives section was restructured accordingly.

Round 3

Reviewer 2 Report

Accept